# Cell fate specification modes shape transcriptome evolution in the highly conserved spiral cleavage

Yan Liang [ID][1,3], Jingcheng Wei[1], Yue Kang[2], Allan M Carrillo-Baltodano[1] & José M Martín-Durán [ID][1✉]

## Abstract

Early animal development can be remarkably variable, influenced by lineage-specific reproductive strategies and adaptations. Yet, early embryogenesis is also strikingly conserved in certain groups, such as Spiralia. In this clade, a shared cleavage program (i.e., spiral cleavage) and similar cell lineages are ancestral to at least seven phyla. Why early development is so conserved in specific groups and plastic in others is not fully understood. Here, we investigated two annelid species (*Owenia fusiformis* and *Capitella teleta*) with spiral cleavage but different modes of specifying their primary progenitor cells. By generating high-resolution transcriptomic time courses from the oocyte to gastrulation, we demonstrate that transcriptional dynamics differ markedly between these species during spiral cleavage and instead reflect their distinct timings of embryonic organiser specification. However, the end of cleavage and gastrulation exhibit high transcriptomic similarity, when orthologous transcription factors share gene expression domains, suggesting this period is a previously overlooked mid-developmental transition in annelid embryogenesis. Together, our data reveal hidden transcriptomic plasticity during spiral cleavage, indicating an evolutionary decoupling of morphological and transcriptomic conservation during early embryogenesis.

**Keywords** Spiral Cleavage; Maternal-to-Zygotic Transition; Zygotic Genome Activation; Phylotypic Stage; *Annelida*
**Subject Categories** Development; Evolution & Ecology

## Introduction

Upon fertilisation, animal zygotes enter a phase of intense cell division called cleavage, resulting in an embryo––a blastula––made of developmentally committed cells that segregate into the main germ layers during gastrulation (Barresi and Gilbert, 2023; Wolpert et al, 2019). As a foundational stage for subsequent development, broadly conserved cleavage modes are recognised in animals (Barresi and Gilbert, 2023; Wolpert et al, 2019). In the holoblastic radial cleavage, the zygote and its daughter cells divide completely and maintain a radially symmetrical arrangement, at least during the early rounds of cell division. This cleavage mode is widespread, observed throughout the animal phylogeny, and is likely ancestral (Valentine, 1997). However, specific clades, such as tunicate chordates and acoel flatworms, have evolved idiosyncratic and highly stereotypic cleavage programmes, like bilateral cleavage and duet spiral cleavage, respectively (Kimura et al, 2021; Lemaire et al, 2008). Indeed, the initial steps of animal embryogenesis can vary greatly, often influenced by the reproductive strategies of each animal lineage. This is most obvious in animals whose eggs have large amounts of yolk, such as fish, birds, insects, and cephalopods (Barresi and Gilbert, 2023; Wolpert et al, 2019), where complete cytokinesis during cell division does not occur. Deviations in early cleavage can also be more subtle. For example, although sea stars and sea urchins share holoblastic radial cleavage, only the latter form their larval skeleton from a set of vegetal micromeres, a unique trait of this echinoderm lineage (Emura and Yajima, 2022). How do distinct cleavage modes evolve? How do novelties appear during early embryogenesis, and by what genetic and developmental mechanisms?

Spiral cleavage is an ancient and highly conserved early developmental programme found in at least seven major animal groups within Spiralia (or Lophotrochozoa), one of the three largest branches of bilaterally symmetrical animals (Laumer et al, 2015; Marletaz et al, 2019; Struck et al, 2014). The alternating shift of the mitotic spindle along the animal-vegetal embryonic axis from the 8-cell stage onwards characterises this cleavage mode and results in a spiral-like arrangement of the animal-pole blastomeres when viewed from above, hence the name of this early embryogenesis (Hejnol, 2010; Henry, 2014; Martin-Duran and Marletaz, 2020). In addition to the conserved pattern of cell divisions, embryos with spiral cleavage also exhibit broadly conserved cell lineages (Nielsen, 2004, 2005). Thereby, equivalent blastomeres in closely and more distantly phylogenetically related species often act as progenitors of similar cell types, tissues, and organs. Despite the conservation of cleavage patterns and cell lineages, embryos with spiral cleavage exhibit two markedly different strategies for specifying their primary cell lineages and establishing their axial patterning (Fig. 1) (Hejnol, 2010; Henry, 2014; Lambert, 2010; Martin-Duran and Marletaz, 2020). In the so-called equal (or conditional) spiral cleavage, bilateral symmetry is established with the inductive specification of a blastomere––the so-called 4d micromere–– that

[1]School of Biological and Behavioural Sciences, Queen Mary University of London, Mile End Road, E1 4NS London, UK. [2]Imperial College of London, Department of Life Sciences, Exhibition Road, SW7 2AZ London, UK. [3]Present address: Wellcome Sanger Institute, Wellcome Genome Campus, Hinxton. CB10 1SA, Cambridge, UK. ✉E-mail: chema.martin@qmul.ac.uk

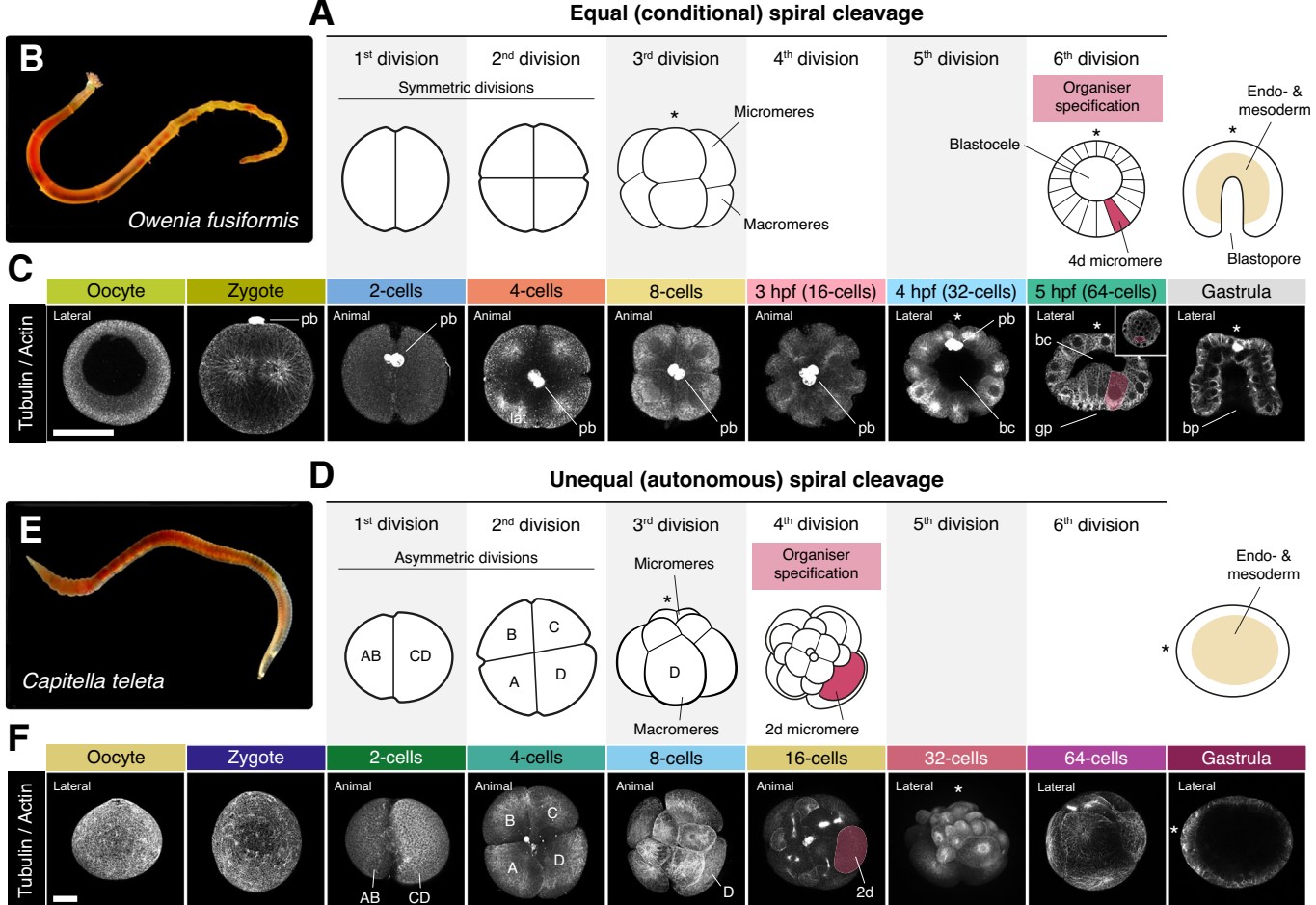

**Figure 1. Equal and unequal spiral cleavage in two annelid species.**

(A, D) Schematic drawings of equal (or conditional) (A) and unequal (or autonomous) (D) spiral cleavage as exemplified by the early development of the annelids *O. fusiformis* (B) and *C. teleta* (E). The drawings highlight the two different types of first zygotic divisions, the idiosyncratic spiral arrangement of the animal and vegetal blastomeres at the 8-cell stage, the different timings of embryonic organiser specification, and distinct gastrula morphologies in embryos that will develop into planktotrophic (*O. fusiformis*) and lecithotrophic (*C. teleta*) larvae. (C, F) Z-projections of confocal stacks of oocytes and embryos at the exact stages collected for transcriptomic profiling in *O. fusiformis* (C) and *C. teleta* (F). Samples are stained against tubulin and actin (both in grey) to reveal cell membranes and contours. In *C. teleta* (F), the blastomere lineages at the 2-, 4- and 8-cell stages are highlighted. The cell acting as an embryonic organiser (the 4d micromere in *O. fusiformis* and the 2d blastomere in *C. teleta*) is false-coloured in red at the 5 hpf and 16-cell stage, respectively. Drawings are not to scale, and asterisks indicate the animal/anterior pole. bc blastocoele, bp blastopore, hpf hours post-fertilisation, pb polar bodies. In (C, F), scale bars are 50 μm. (C, F) All images come from representative specimens of at least two biological replicates. (E) Copyright 2025 by François Michonneau (CC-BY 4.0). Source data are available online for this figure.

acts as an embryonic organiser at the fifth or sixth round of cell division (the 32- or 64-cell stages, depending on the species) (Fig. 1A–C) (Lambert, 2010). In molluscs and annelids, the FGF receptor pathway and the ERK1/2 transducing cascade regulate this process (Hejnol, 2010; Seudre et al, 2022a; Tan et al, 2022, 2023). However, symmetry breaking occurs much earlier in many molluscan and annelid embryos. In these species, spiral cleavage is regarded as unequal (or autonomous) because the asymmetric segregation of maternal determinants into a larger cell by the second round of cell division (the 4-cell stage) defines the posterodorsal fate and the progenitor lineage of the embryonic organiser (Fig. 1D–F) (Freeman and Lundelius, 1992; Render, 1989). While equal spiral cleavage occurs in all major animal clades with spiral cleavage and is considered the ancestral condition, the unequal mode has evolved independently multiple times (Freeman

and Lundelius, 1992). Therefore, spiral cleavage, with overall conserved cell division patterns and cell lineages but recursively evolved distinct modes of cell fate specification and axial patterning, is an ideal system to explore the developmental and evolutionary mechanisms that generate phenotypic change during early animal embryogenesis.

Here, we investigate the genome-wide impact of conditional and autonomous cell fate specification strategies on spiral cleavage. Does the conservation of an ancient programme of cell division patterns and cell lineages constrain the unfolding of developmental programmes during spiral cleavage, or do differences in cell fate specification modes play a more prominent role in defining gene expression dynamics? To answer these, we generated high-resolution transcriptomic profiles at equivalent time points during the early embryogenesis of *Owenia fusiformis* and *Capitella teleta*,

two annelid worms with equal/conditional and unequal/autonomous spiral cleavage, respectively (Fig. 1) (Amiel et al, 2013; Carrillo-Baltodano et al, 2021; Seudre et al, 2022a). Our findings reveal that these two annelids undergo roughly similar transcriptomic transitional phases during spiral cleavage, with maternal genes likely decaying around the 16-cell stage, and zygotic genome activation starting as early as the 4-cell stage. Nonetheless, the genes and temporal dynamics defining some of these phases are markedly different and mirror the timing and mechanistic differences in axial patterning and embryonic organiser specification between O. fusiformis and C. teleta (Amiel et al, 2013; Seudre et al, 2022a). Despite these differences, the embryos of these annelids exhibit a period of maximal similarity––at the transcriptomic and molecular patterning level––at the late cleavage and gastrula stage, suggesting that, unlike previous hypotheses (Levin et al, 2016; Paps et al, 2015; Wu et al, 2019; Xu et al, 2016), spiral-cleaving species exhibit a mid-developmental transition, or phylotypic stage, during embryogenesis. Together, our study uncovers an unexpected transcriptomic diversity between species with an otherwise broadly conserved cleavage programme, suggesting that distinct cell-fate specification strategies outweigh the conservation of cleavage patterns and overall cell lineages in the evolution of developmental programmes in Spiralia.

# Results

## Global transcriptional dynamics during spiral cleavage

To investigate the global transcriptional dynamics during annelid spiral cleavage, we generated high-resolution time courses of bulk RNA-seq data for the annelids O. fusiformis and C. teleta, collecting samples in biological duplicates of active or mature oocytes, zygotes, and at each round of cell division until the gastrula stages (Fig. 1C,F). In O. fusiformis, given the small size of the embryos, the 16-, 32-, and 64-cell stages were collected based on developmental timing (3-, 4-, and 5-h post-fertilisation [hpf], respectively), following a previous description of its early embryogenesis (Carrillo-Baltodano et al, 2021). In all cases, replicates correlated highly (Appendix Figs. S1A,B and S2A,B), with developmental timing accounting for most of the variance (62.4 and 57.6% for O. fusiformis and C. teleta, respectively) in both species (Fig. 2A,B). However, unlike the activated oocyte of O. fusiformis, the oocyte stage of C. teleta was markedly transcriptionally distinct from the fertilised zygote (Fig. 2B; Appendix Fig. 2A), suggesting that either the collected oocytes were not fully mature or that significant transcriptomic changes occur during mating and fertilisation in this species. Since fertilisation is poorly understood in C. teleta and cannot be experimentally controlled (Blake et al, 2009; Reish, 1974), we could not discern between these two possibilities. Thus, we discarded the oocyte stage for this species in the downstream analyses.

Similarity clustering revealed three transcriptionally distinct groups during spiral cleavage in the two annelids (Fig. 2C). The earliest cluster comprises the oocyte (in O. fusiformis) and early cleavage stages up to the 8-cell stage. The late cleavage time points (3 to 5 hpf in O. fusiformis and the 16-cell to 64-cell stages in C. teleta) and the gastrula stages make up the two other groups. Accordingly, the number of expressed genes (those with transcripts

per million [TPM] values above 2; Appendix Figs. S1C and S2C) is relatively constant between stages in the first cluster in both species (Fig. 2D). However, there is a continuous increase in the number of expressed genes during the late cleavage and, particularly, with gastrulation. This increase in C. teleta starts as early as the 16-cell stage and occurs more sharply than in O. fusiformis, in which the rise is gradual and not evident until the 64-cell stage (Fig. 2D). Notably, these distinct dynamics mirror the different timings of axial specification in O. fusiformis and C. teleta (Figs. 1A,D and 2C) (Amiel et al, 2013; Seudre et al, 2022a), suggesting that conditional and autonomous specification of the embryonic organiser influence global transcriptional trends during cleavage in these annelids.

Statistically significant changes in gene expression support the distinct transcriptional activation dynamics between O. fusiformis and C. teleta (Fig. 2E,F). While there is a gradual increase in the differentially expressed genes (DEGs) becoming upregulated from 3 hpf onwards in O. fusiformis, the number of upregulated DEGs at the equivalent 16-cell stage is more prominent, and a steep change from the levels of DEGs in the previous pairwise comparison in C. teleta (78 in O. fusiformis vs 1890 in C. teleta) (Appendix Table S1; Datasets EV1–EV30). This agrees with the timing of specification of the embryonic organiser in C. teleta and supports its role in triggering transcriptional programmes involved in embryonic patterning and cell fate commitment at that stage in this species (Amiel et al, 2013). In both species, however, gastrulation involves a prominent transcriptional reshaping, with five to seven times more up- and downregulated DEGs at that stage compared with previous cleavage stages (Fig. 2E,F). Accordingly, Gene Ontology (GO) terms enriched in the genes upregulated in the gastrula of both species are involved in development and transcriptional regulation (Appendix Figs. S3H and S5G). Interestingly, we identified DEGs between the oocyte and zygote (in O. fusiformis) and the 2-cell stage and zygote (in C. teleta). In the two annelids, downregulated DEGs are more abundant at these stages (290 in O. fusiformis and 1,101 in C. teleta), are involved in metabolism and translation (O. fusiformis) and protein transport and localisation (C. teleta) and might likely represent maternal genes of quick decay (Fig. 2E,F; Appendix Figs. S4A and S6A). Nonetheless, we also detected 203 and 527 upregulated genes in O. fusiformis and C. teleta, respectively, at those stages (Fig. 2E,F). These are enriched for GO terms related to cellular organisation (Appendix Figs. S3A and S5A) and might represent differentially polyadenylated transcripts upon fertilisation and the first zygotic division, as observed in other invertebrates (Rouhana et al, 2023; Wilt, 1973). Altogether, our high-resolution dataset demonstrates that annelid embryos shared similar transcriptomic phases during spiral cleavage. Yet, the modes and timing of organiser specification influence the span and intensity of these phases during early development.

## The maternal-to-zygotic transition in *O. fusiformis* and *C. teleta*

In O. fusiformis and C. teleta, the first prominent increase in DEGs occurs at the 16-cell stage, the fourth round of cell division after fertilisation (Fig. 2E,F). This suggests that zygotic genome activation, and thus the maternal-to-zygotic transition (MZT), might occur around that stage, albeit with different intensities, in the two annelids. To characterise the exact timing of zygotic

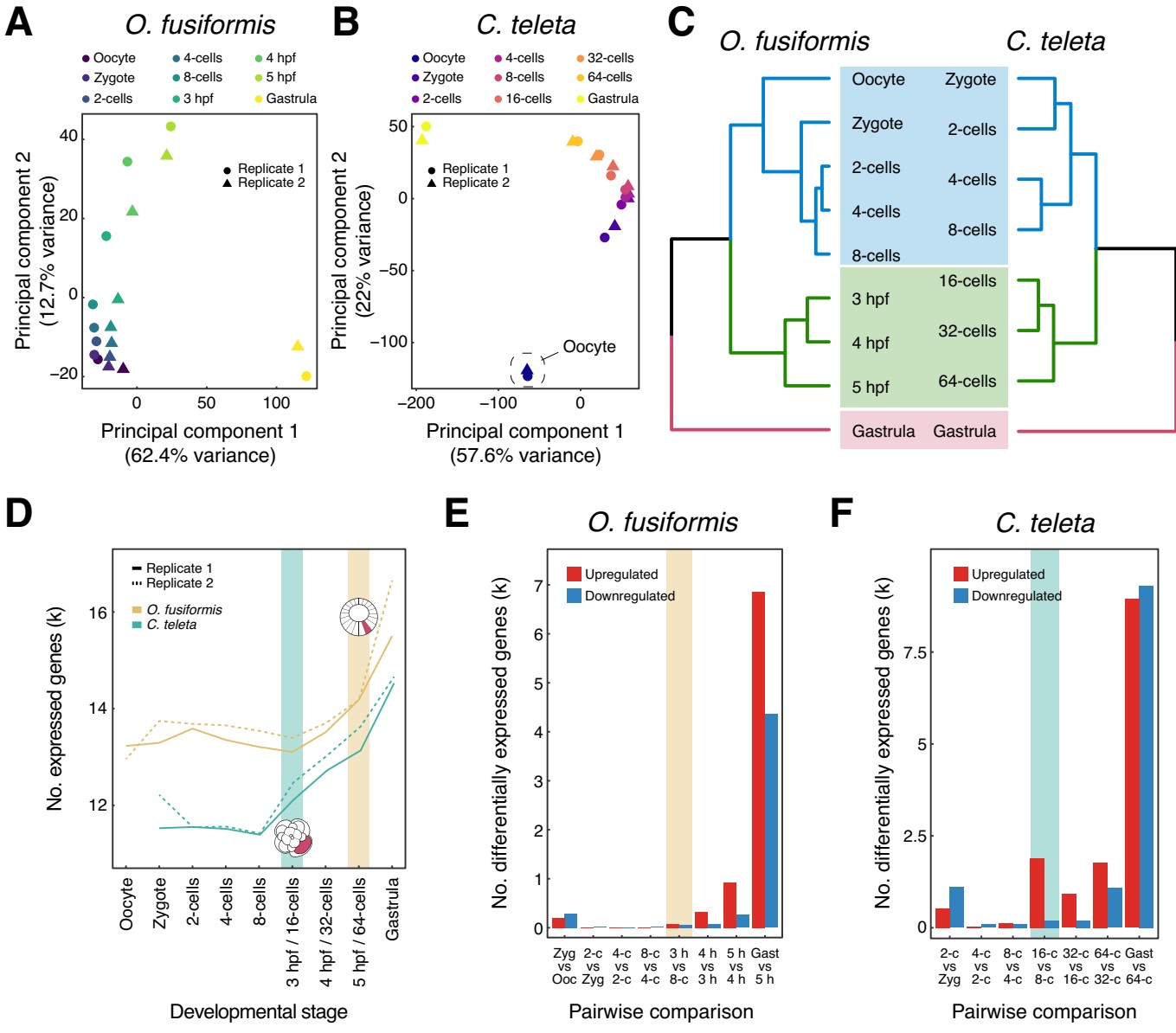

**Figure 2.  Global transcriptional dynamics during conditional and autonomous spiral cleavage.**

(**A, B**) Principal component analyses support that developmental time accounts for the most considerable fraction of the variance in the transcriptomic time courses during spiral cleavage in *O. fusiformis* and *C. teleta*. (**C**) Similarity clustering reveals three primary sample groups during spiral cleavage in both species. (**D**) *O. fusiformis* and *C. teleta* show different dynamics in the total number of expressed genes during spiral cleavage, which align with the different timings of organiser specification in these species (highlighted by a yellow and green bar, respectively). (**E, F**) Bar plots depicting the number of differentially expressed genes (upregulated in red and downregulated in blue) during spiral cleavage in *O. fusiformis* (**E**) and *C. teleta* (**F**). The earliest signs of zygotic genome activation are highlighted with a yellow and green bar, respectively.

genome activation, we treated embryos of *O. fusiformis* and *C. teleta* with actinomycin D, a commonly used transcriptional inhibitor (Sobell, 1985), in staggered time windows from the 2-cell stage until 6 hpf in *O. fusiformis* and the 32-cell stage in *C. teleta* (Fig. 3A,B). In both species, blocking zygotic transcription until the 8-cell stage does not compromise normal development, although the embryos of *O. fusiformis* look smaller (Fig. 3A,B). However, if zygotic transcription is inhibited until the 16-cell stage or later, the embryos develop abnormally (although some cellular differentiation is observed when treating between 3 and 4 hpf in *O.*

*fusiformis*) (Fig. 3A,B), consistent with our RNA-seq data and the timing of the first wave of DGE in these annelids (Fig. 2E,F). Importantly, samples treated until 5 hpf (in *O. fusiformis*) and the 32-cell stage (in *C. teleta*) exhibited more severe phenotypes and failed to gastrulate, supporting that the large increase in zygotic activation occurs earlier in *C. teleta* and coincides with the specification of the embryonic organiser in both annelids (Fig. 3A,B). Immunostaining against RPB1, the largest subunit of DNA polymerase II, indicates that the nuclearisation of the transcriptional machinery starts at the 4-cell stage in both *O.*

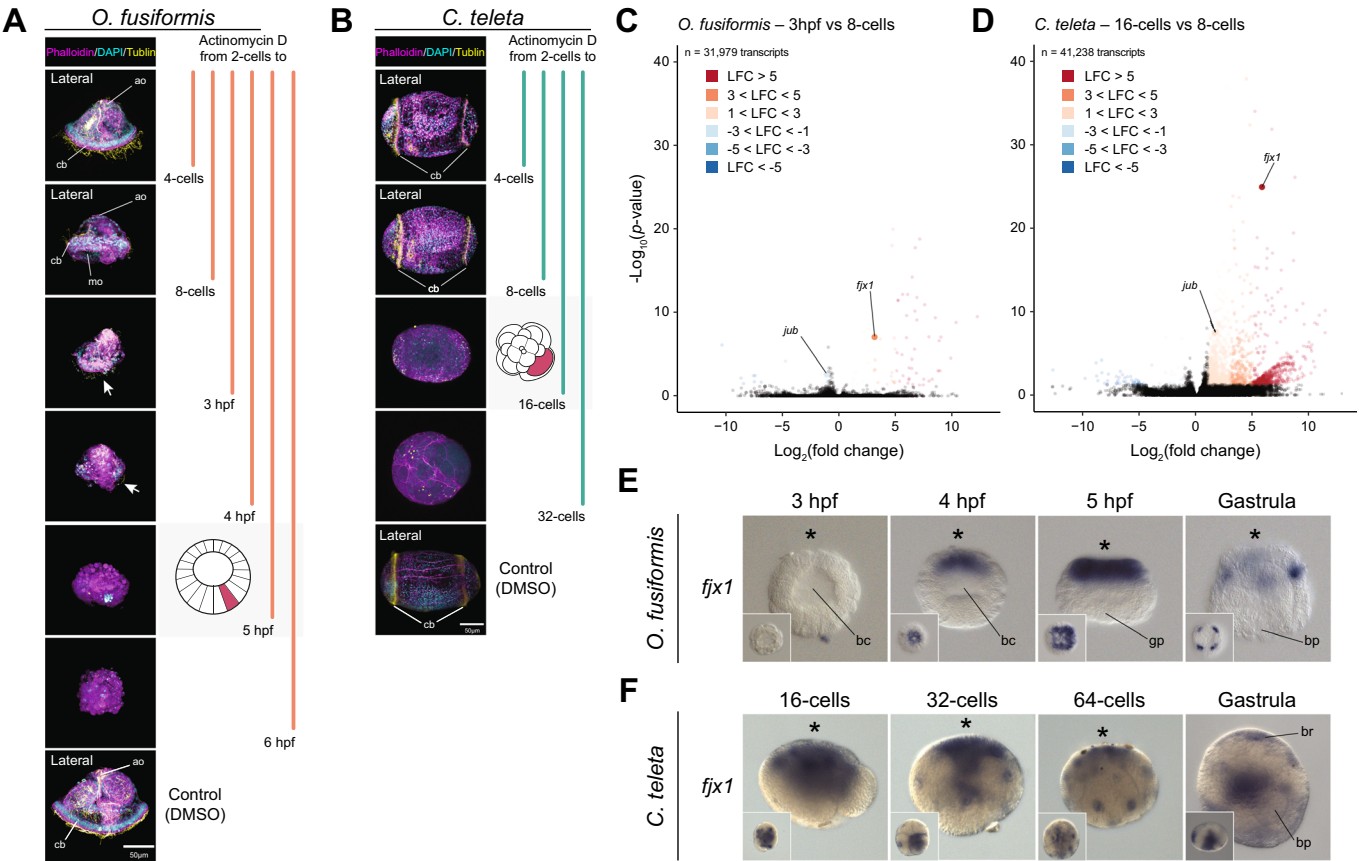

**Figure 3. The zygotic genome activation in *O. fusiformis* and *C. teleta*.**

(A, B) Z-projections of confocal stacks showing the morphological phenotype of 24 h post-fertilisation *O. fusiformis* larvae (A) and stage 5 *C. teleta* larvae (B) after windows of actinomycin D treatment during early cleavage. Inhibition of zygotic transcription from the 16-cell stage on compromises normal embryogenesis in the two annelids. However, in *O. fusiformis*, there is cellular differentiation between 3 to 5 hpf (white arrows), and in both annelids, the most severe phenotypes appear when inhibiting zygotic genome activation during the specification of the embryonic organiser and major wave of differential gene expression (grey background). (C, D) Volcano plots indicating up- and downregulated genes in the 8-cell to 16-cell transition in *O. fusiformis* (C) and *C. teleta* (D). Only two one-to-one orthologs (*fjx1* and *jub*) are differentially expressed at this stage in both species. LFC stands for log-fold change. *P*-values in (C, D) were derived from the described DESeq2 pipeline (Wald test) and Benjamini–Hochberg-adjusted. (E, F) Whole-mount in situ hybridisation of *jbx1* from the 16-cell stage (or 3 hpf in *O. fusiformis*) until gastrulation in *O. fusiformis* (E) and *C. teleta* (F). In *O. fusiformis*, *jbx1* is asymmetrically expressed in the anterior ectoderm. In *C. teleta*, *jbx1* is broadly expressed during cleavage and asymmetrically localised in the anterior neuroectoderm and endoderm in the gastrula. All images come from representative specimens of at least two biological replicates. Asterisks in (E, F) indicate the animal pole. ao apical organ, bc blastocoele, bp blastopore, br brain, cb ciliary band, gp gastral plate, mo mouth. Source data are available online for this figure.

*fusiformis* and *C. teleta* (Fig. EV1A,C), despite *rpb1* being a maternally provided transcript (Fig. EV1B,D). Interestingly, RPB1 exhibits an asymmetric nuclear signal at the 4-cell stage in *C. teleta*, with blastomeres C and D displaying a stronger signal than A and B (Fig. EV1C). Therefore, the timing of zygotic genome activation is equivalent in annelids with conditional and autonomous spiral cleavage, likely occurring between the 4- and the 16-cell stages in *O. fusiformis* and *C. teleta*.

To explore the genes potentially involved in the zygotic genome activation and how this process compares between these species, we used previously calculated functional annotations and one-to-one orthologs in *O. fusiformis* and *C. teleta* (Martin-Zamora et al, 2023). Upregulated genes at 3 hpf (in *O. fusiformis*) and the 16-cell stage (in *C. teleta*) are enriched for GO categories related to transcription (Fig. 3C,D; Appendix Figs. S3E and S5D). In *O. fusiformis*, these involve over ten unclassified TALE homeoboxes, which are expanded in Spiralia and expressed during early cleavage

along the animal-vegetal axis (Morino et al, 2017), and several Fox genes, including many of the expanded *foxQ2* genes that are asymmetrically expressed in the animal pole at 3 hpf (Seudre et al, 2022b) and *foxA*, which is a well-known pioneer factor initiating chromatin opening (Barral and Zaret, 2024; Cirillo et al, 2002). As expected by their higher number, upregulated genes at the 16-cell stage are diverse and involved in numerous developmental processes in *C. teleta*. These include, among others, genes directly involved in transcriptional regulation (e.g. transcription elongation factors and histone demethylases), cell-fate specification transcription factors (e.g. *Six* genes, *otx*, *otp*, *Fox* genes) and signalling pathway components (e.g. Wnt ligands, Frizzled and FGF receptors and TGF-b modulators) (Dataset EV23). However, only two genes upregulated at the 16-cell stage in *C. teleta* had a differentially expressed ortholog at 3 hpf in *O. fusiformis* (Fig. 3C,D). One––*fjx1*, the homologue of *four jointed* in *Drosophila melanogaster*, involved in conferring positional information as a potential downstream

target of the Notch Delta pathway (Buckles et al, 2001; Rock et al, 2005)––was upregulated in both species, but the other one––*jub*, a gene encoding for AJUBA, a protein involved in many developmental processes, from transcriptional co-repression to the negative regulation of the Hippo signalling (Schimizzi and Longmore, 2015)––was downregulated in *O. fusiformis* (Fig. 3C). Therefore, although MZT may occur at similar developmental stages, the first signs of zygotic genome activation largely involve different genes and regulatory outcomes in *O. fusiformis* and *C. teleta*.

Given the shared upregulation of *fjx1* at the 16-cell stage, we compared the expression of this gene from this time point until gastrulation in *O. fusiformis* and *C. teleta* (Fig. 3E,F). Consistent with its modest upregulation (Fig. 3C), we did not detect expression for *fjx1* by whole-mount in situ hybridisation at 3 hpf in *O. fusiformis* (Fig. 3E). At 4 and 5 hpf, *fjx1* was, however, strongly expressed in the animal-most ectoderm but not in the top micromeres, becoming restricted to four radially symmetrical cell clusters in the animal hemisphere at the gastrula stage (Fig. 3E). This pattern is reminiscent of the expression of several *foxQ2* genes during spiral cleavage in *O. fusiformis*, which are already expressed at the 16-cell stage (Seudre et al, 2022b). In *C. teleta*, however, *fjx1* showed a broader expression domain from the 16- to the 64-cell stage, and its expression became restricted to the anterior neuroectoderm and endoderm at the gastrula stage (Fig. 3F). Therefore, orthologous genes also follow different spatial expression dynamics upon zygotic genome activation, which occurs asymmetrically throughout the embryo, at least in *O. fusiformis*.

## Gene-specific transcriptional dynamics differ in spiral cleavage

Although spiral cleavage is a highly stereotypical developmental programme (Hejnol, 2010; Henry, 2014; Martin-Duran and Marletaz, 2020; Nielsen, 2004, 2005), our findings at the 16-cell stage hint at gene-specific differences between annelid embryos at similar developmental time points (Fig. 3C,D). To attain a comparative, genome-wide view of gene expression dynamics between *O. fusiformis* and *C. teleta*, we first used one-to-one orthologs ($n = 7607$) to calculate the Jensen-Shannon index of transcriptomic similarity during early development in the two annelids (Fig. 4A). Consistent with our observations at the 16-cell stage, transcriptomic similarity is low during cleavage stages but increases in the late blastula and gastrula stages (Fig. 4A). Notably, a similar trend is observed when comparing *O. fusiformis* and *C. teleta* with other annelids that have available transcriptomic data during spiral cleavage (Fig. EV2A,B), despite technical and sampling differences between these datasets. Soft *k*-means clustering of all expressed genes during early embryogenesis (31,323 for *O. fusiformis* and 38,662 for *C. teleta*) revealed five and seven clusters of temporally coexpressed genes in *O. fusiformis* and *C. teleta*, respectively (Fig. 4B,C; Datasets EV31 and EV32). The first two temporally active clusters in both species are maternal genes that are either oocyte-/zygote-specific (cluster 1; 3593 genes in *O. fusiformis* and 2934 in *C. teleta*) and have a different codon usage than zygotically expressed genes (Appendix Fig. S7A,C,D) or that peak shortly after fertilisation (cluster 2; 5317 genes in *O. fusiformis* and 5450 in *C. teleta*) (Fig. 4B,C). Genes in cluster 1 get quickly removed with fertilisation and the first zygotic division, while genes in cluster 2 decay before the 8-cell stage in both species. GO and

KEGG pathway enrichment analyses indicate that these clusters are overrepresented by genes involved in metabolism, suggesting an early clearance of transcripts that may be involved in oocyte maturation (Appendix Figs. S8A–F and S9A–F). In *O. fusiformis*, 168 of the 290 differentially downregulated genes between the oocyte and the zygote (57.93%; Fig. 2D) are oocyte-specific (cluster 1), supporting a rapid elimination of transcripts potentially involved in oocyte maturation upon fertilisation (Appendix Fig. S4A). Likewise, 162 of the 203 upregulated genes with fertilisation (57.93%; Fig. 2D) in *O. fusiformis*, and potentially polyadenylated, were not cleared during early cleavage (i.e. were not in clusters 1 and 2), suggesting that, as in other organisms, polyadenylation might be a mechanism to differentially stabilise maternally deposited transcripts (Rouhana et al, 2023; Wilt, 1973).

In both species, the remaining clusters (3 to 5 in *O. fusiformis* and 3 to 7 in *C. teleta*) comprise genes that increase in expression around the 8-cell stage or later (Fig. 4B,C). In *O. fusiformis*, early zygotic genes (cluster 3; 4610 genes) become more highly expressed at the 8-cell stage but are primarily restricted to the 3- and 4-hpf stages. This early upregulation is consistent with a nuclearisation of the RNA polymerase II at the 4-cell stage (Fig. EV1). These genes are still involved in housekeeping functions, such as metabolism, cellular transport, and protein/organelle localisation (Fig. 4B; Appendix Fig. S8G–I). In contrast, late zygotic genes (cluster 4; 4109 genes) peak at 5 hpf and are enriched in GO terms involved in MAPK signalling, amongst other signalling pathways (Appendix Fig. S8J–L), which is consistent with the activation of ERK1/2 at this stage to define the embryonic organiser in *O. fusiformis* (Seudre et al, 2022a). Lastly, a large (13,695 genes) gene set enriched in GO terms related to animal development (cluster 5) emerges upon gastrulation in *O. fusiformis* (Fig. 4B; Appendix Fig. S8M–O). *Capitella teleta* also showed a gene set (cluster 3; 6161 genes) of early zygotic expression involved in metabolism, cellular transport, and protein localisation that is lowly expressed in the zygote and gradually increases its expression to peak at the 8-cell stage, again consistent with the dynamics of RNA polymerase II immunostaining at these stages (Fig. 4C; Appendix Fig. S9G–I). Likewise, this species also exhibits a large gene cluster restricted to gastrulation (cluster 7; 13,541 genes; Fig. 4C). However, unlike *O. fusiformis*, *C. teleta* exhibits stage-specific clusters of temporally coregulated genes for the 16-cell (2310 genes), 32-cell (3104 genes), and 64-cell (4841 genes) stages (Fig. 4C). Genes in these clusters involve diverse biological processes, but DNA metabolism and GPCR-mediated neuropeptide signalling are common (Appendix Fig. S9J–R). However, around half of the genes in these clusters do not have a functional annotation (1040 genes or 45% in cluster 4; 1565 or 50.42% in cluster 5; and 2161 genes or 44.64% in cluster 6), indicating that lineage-restricted genes might play a role at these cleavage stages in *C. teleta*. Altogether, our findings support that *O. fusiformis* and *C. teleta* generally follow similar transcriptional trends. However, *C. teleta* shows more complex and dynamic gene coexpression patterns of zygotically expressed genes, whereas *O. fusiformis* mainly restricts zygotic expression with and after the specification of the embryonic organiser at 5 hpf.

Despite the shared transcriptional trends between *O. fusiformis* and *C. teleta* (Fig. 4B,C), the gene-specific transcriptional similarity is low during spiral cleavage (Fig. 4A). This might indicate that either cluster composition is dissimilar, or similar genes comprise equivalent clusters but are expressed at different levels. To test this,

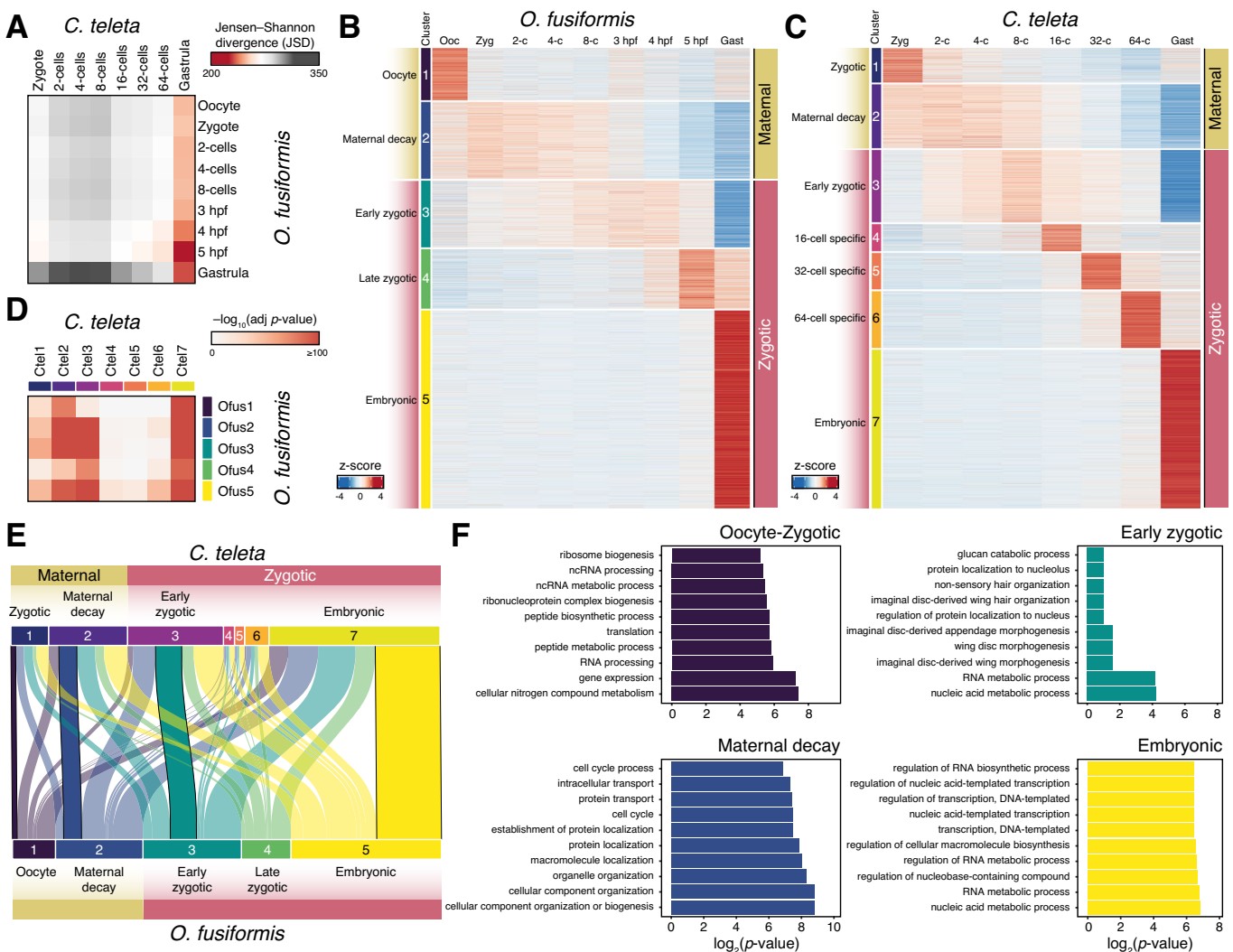

**Figure 4. Distinct transcriptomic routes during spiral cleavage in the annelids O. fusiformis and C. teleta.**

(A) Jensen-Shannon distance represents the transcriptomic divergence during the spiral cleavage of *O. fusiformis* and *C. teleta*. The point of maximal transcriptomic similarity between these annelids occurs at the late cleavage and gastrulation. (B, C) Soft *k*-means clustering of temporally coexpressed genes in *O. fusiformis* (B) and *C. teleta* (C). In both species, the first two clusters likely represent maternal transcripts that decay early, while the rest largely or entirely comprise zygotically expressed genes. Unlike *O. fusiformis*, *C. teleta* has cleavage-specific clusters at the 16-cell, 32-cell and 64-cell stages. (D) Comparison of gene family cluster composition between *O. fusiformis* and *C. teleta*. Although gene-specific transcriptional dynamics are dissimilar, the deployment of gene families in maternal, early zygotic, and embryonic clusters is similar between these annelids. *P* values were derived from upper-tail hypergeometric tests and Benjamini–Hochberg-adjusted (adj. *P* value). (E) Alluvial plot depicting the comparative deployment of one-to-one orthologs (*n* = 7607) in clusters of temporally coexpressed genes during spiral cleavage in *O. fusiformis* and *C. teleta*. The embryonic clusters exhibit the most extensive conservation of gene composition between species. (F) Gene Ontology (GO) enrichment of shared orthologs between oocyte/zygote, maternal decay, early zygotic, and embryonic clusters, according to the GO annotation of *O. fusiformis*. While oocyte/zygotic genes are involved in metabolism, shared orthologs of maternal decay are involved in the cell cycle and cellular organisation, and early zygotic and embryonic genes are involved in transcriptional control and development. *P* values were computed from upper-tail Fisher's exact tests to detect overrepresented terms.

we first performed cross-species pairwise comparisons of orthogroup composition between clusters of temporally coexpressed genes (Fig. 4D). Clusters comprising genes of maternal decay (clusters 2), early zygotic expression (clusters 3), and embryonic expression (cluster 5 in *O. fusiformis* and cluster 7 in *C. teleta*) showed high inter-species conservation, supporting these include gene families involved in core, conserved embryonic processes, such as early cell cycle progression, and body patterning (Appendix Figs. S8D–I,M–O and S9D–I,S–U). However, *C. teleta*'s cleavage-specific clusters showed little correspondence with any cluster in *O.*

*fusiformis* (Fig. 4D). Moreover, embryonic clusters in both species, especially in *C. teleta*, showed similarity with earlier gene sets of coexpressed genes (Fig. 4D). Next, we compared the cluster allocation of one-to-one orthologs (Fig. 4E). Differences in orthogroup composition are general and profound (Figs. 4E and EV3A). For example, 8.18% (622) of embryonic genes in *O. fusiformis* exhibited maternal expression in *C. teleta*; conversely, 11.29% (859) of embryonic genes in *C. teleta* were maternal in *O. fusiformis*. However, as expected, embryonic (14.92% in *O. fusiformis*), early zygotic (6.02% in *O. fusiformis*), and maternal

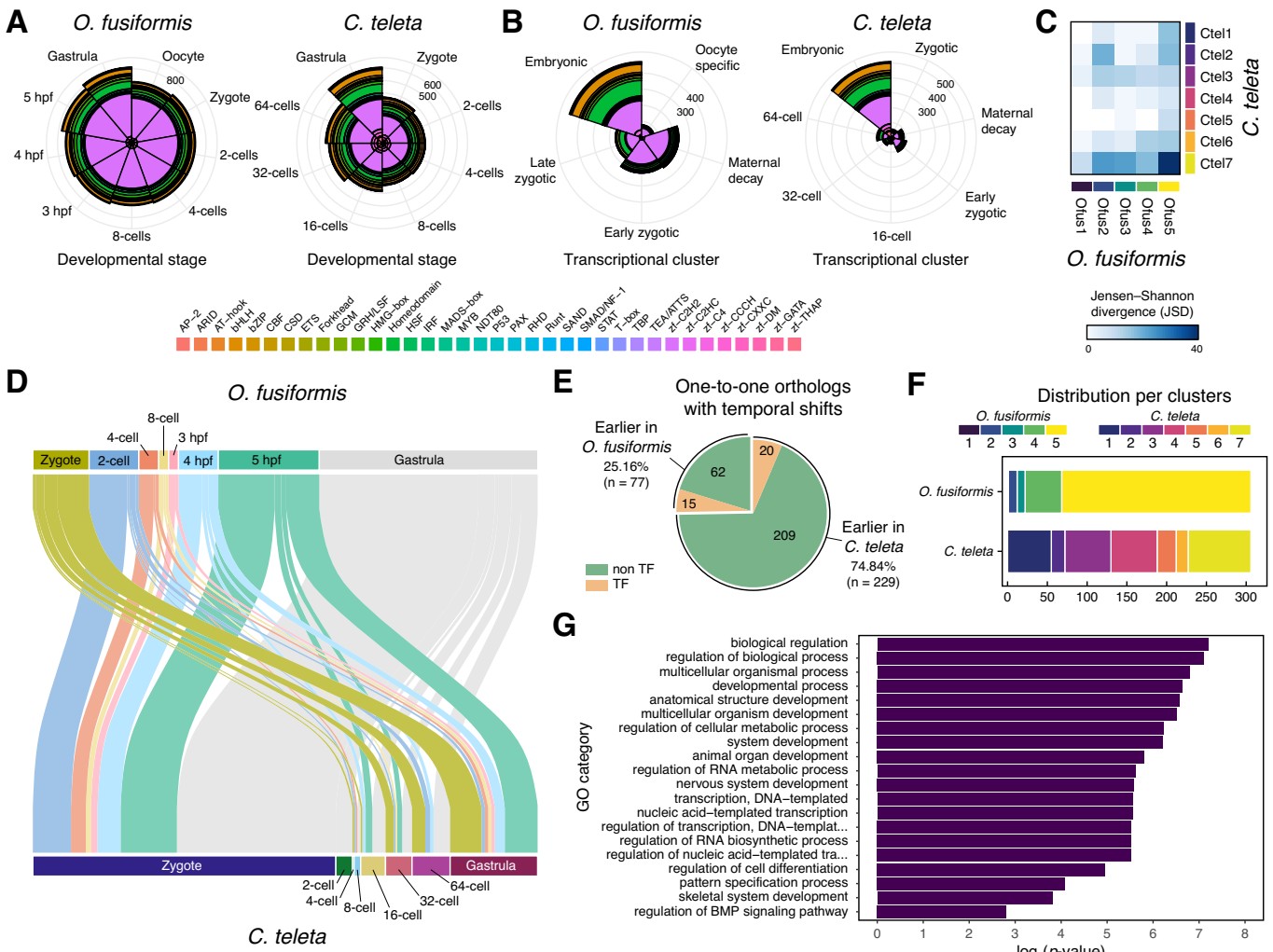

**Figure 5. The comparative dynamics of transcription factor activation in conditional and autonomous annelids.**

(A, B) Nightingale rose charts of transcription factor (TF) distribution according to developmental time points (A) and clusters of coexpressed genes (B). (C) Comparison of gene family cluster composition between *O. fusiformis* and *C. teleta*. TF similarity in clusters of genes showing maternal decay and embryonic expression is high. However, there are prominent shifts in TF gene families allocated to the embryonic cluster in one species to earlier clusters in the other. (D) Alluvial plot depicting the comparative deployment of one-to-one orthologs (n = 306) exhibiting shifts in temporal activation (TPM >2) during spiral cleavage in *O. fusiformis* and *C. teleta*. (E) Pie chart showing that most genes with a temporal shift between *O. fusiformis* and *C. teleta* are expressed earlier in the latter. (F) The bar plots indicate the allocation of genes with a temporal shift to the clusters of coregulated genes in *O. fusiformis* and *C. teleta*. Most genes exhibiting temporal shifts between these species belong to late clusters in *O. fusiformis* and shift towards earlier clusters in *C. teleta*. (G) Gene Ontology (GO) enrichment of one-to-one orthologs (n = 306) exhibiting temporal shifts in transcriptional activation between *O. fusiformis* and *C. teleta*, according to the GO annotation of *C. teleta*. P values were derived from upper-tail Fisher's exact tests.

decay (4.23% in *O. fusiformis*) clusters show the highest (yet still relatively low) proportions of conserved one-to-one orthologs (Fig. EV3A). These shared orthologs are enriched in GO terms for cell-cycle progression and protein/organelle localisation (maternal decay clusters), morphogenesis (early zygotic) and transcription (embryonic) (Figs. 4F and EV3B–F). Although the long evolutionary distance between *O. fusiformis* and *C. teleta* likely influences the extent to which gene-specific transcriptional dynamics differ during cleavage in these species, our findings indicate extensive temporal differences in gene usage between *O. fusiformis* and *C. teleta*. Therefore, only a small proportion of genes might be essential and evolutionarily conserved to maintain the fundamental properties of spiral cleavage and annelid embryogenesis.

## Temporal shifts in the activation of developmental programmes

Because genes conserved between *O. fusiformis* and *C. teleta* are enriched in GO categories associated with morphogenesis and transcription (Fig. 4F), we hypothesised that a fraction of those would be transcription factors (TFs) sustaining conserved developmental programmes in early annelid embryogenesis. We thus first characterised the expression dynamics of TFs in these two annelid species. The number of expressed TFs increases as spiral cleavage proceeds in both species (Fig. 5A; Appendix Fig. S10A). However, TFs are especially abundant in the embryonic cluster of temporally coregulated genes (Fig. 5B; Appendix Fig. S10B), as

expected for this gene set to be involved in transcription and activation of body patterning developmental programmes. As per TF class, zinc fingers of class C2H2 dominate the repertoire of expressed TFs (e.g. 454 and 178 at the gastrula stage and embryonic cluster in *O. fusiformis*, respectively; 235 and 187 at the gastrula stage and embryonic cluster in *C. teleta*, respectively), followed by homeodomain-containing and bHLH genes (Datasets EV33–EV36). To further understand how similar TF expression dynamics are between species, we compared the TF gene family composition of clusters of coexpressed genes (Fig. 5C). As with genome-wide approaches, inter-species similarity in TF expression is the highest in embryonic and maternal decay clusters. However, TF gene family composition was also similar between the embryonic cluster in *O. fusiformis* and the zygotic/maternal decay cluster in *C. teleta* and between the embryonic cluster in *C. teleta* and the maternal decay/early zygotic cluster in *O. fusiformis* (Fig. 5C). Therefore, the similarity in gene family usage at equivalent timings of development between these annelids is not only driven by the shared use of structural genes that are essential for animal cleavage (e.g. cell cycle and cytoskeleton-related genes) but also by similar transcription factors.

Despite the overall similarities in TF usage in early and late cleavage, genome-wide and TF-only inter-species cluster composition comparisons also support differential temporal activation of orthologous gene families during spiral cleavage in *O. fusiformis* and *C. teleta* (Figs. 4D and 5C). To identify the exact genes exhibiting these temporal shifts in expression, we compared the activation timing (TPM >2) of one-to-one orthologs during spiral cleavage in the two annelids (Fig. 5D). Consistent with inter-cluster comparisons (Figs. 4D and 5C), the majority of orthologs exhibit a shift from gastrula and late spiral cleavage (5 hpf) in *O. fusiformis* to early cleavage stages, including the zygote, in *C. teleta* (Fig. 5D). Indeed, temporal shifts in gene expression from late stages in *O. fusiformis* to earlier stages in *C. teleta* are more than twice as common as in the other direction (Fig. 5F). Most (75.98%) of the genes–– including 17 out of the 20 TFs (85%)––exhibiting temporal shifts in *O. fusiformis* belong to clusters 4 (late zygotic, with a peak of expression at 5 hpf) and 5 (embryonic, peaking at gastrulation) (Fig. 5E,F; Datasets EV37 and EV38). However, in *C. teleta*, the genes with heterochronic shifts exhibit more diverse temporal dynamics, with more than half belonging to the early clusters 1 to 4 (zygote-specific, maternal decay, early zygotic and 16-cell specific) (Fig. 5F). Generally, the genes exhibiting temporal shifts in transcriptional activation between these species are enriched in GO terms related to development, cell differentiation, transcriptional regulation, and intercellular signalling (e.g. BMP) (Fig. 5G). Maternal genes in *O. fusiformis* that are expressed later during cleavage in *C. teleta* are enriched in GO terms related to ERK1/2 regulation, neuronal development, and DNA methylation, among others (Appendix Fig. S11). Inversely, maternal genes in *C. teleta* that are expressed later in *O. fusiformis* are enriched in GO categories associated with diverse processes, including eye development, the Notch, JNK, and MAPK pathways, as well as cell adhesion and migration (Appendix Fig. S12). Our comparative analyses have thus identified a relatively reduced gene set (306 orthologs, including 35 DNA-binding genes) in *O. fusiformis* and *C. teleta* for future functional investigations, as they might be involved in the early developmental differences in axial and cell fate specification between these two annelids.

To investigate whether similar temporal shifts in gene expression are observed when considering other spiralian species, we compared the high-resolution developmental transcriptomes of *O. fusiformis* and *C. teleta* with the annelids *Platynereis dumerilii* (unequal/autonomous) and *Urechis unicinctus* (equal/conditional), and the mollusc *Crassostrea gigas* (unequal/autonomous) (Fig. EV4). The datasets available for these species covered several cleavage stages, allowing us to identify temporal differences in the activation of one-to-one orthologs during early spiralian development between pairs of species. A larger fraction of genes (between 55.26 and 58.73%) exhibits a predisplacement in their timing of activation between 5 hpf and gastrula stages in *O. fusiformis* and earlier cleavage stages in *P. dumerilii* and *C. gigas*, the two species with unequal development (Fig. EV4A,E). Interestingly, however, this also occurs between *O. fusiformis* and the annelid with equal/conditional spiral cleavage *U. unicinctus* (Fig. EV4B), supporting the observation that transcriptional dynamics during spiral cleavage are inherently plastic across species and evolutionary times (Fig. EV2A). Consistent with our expectation from the comparison with *O. fusiformis* (Fig. 5D), *C. teleta* showed an inverse trend, with only 11.74–26.86% of the genes shifting from a late activation (64-cell stage and gastrula) to earlier time points in the three species (Fig. EV4C,D,F). Notably, the genes exhibiting heterochronic shifts during spiral cleavage in all these comparisons are enriched in GO categories related to organismal development, transcription, morphogenesis and neurogenesis, among others (Appendix Fig. S13). Despite potential confounding factors associated with technical (e.g. number of replicates and sequencing strategy) and sampling (e.g. collecting by time point rather than developmental stage) differences, these broader evolutionary comparisons thus confirm the temporal shifts in the timing of activation of developmental regulators that are generally expressed at late cleavage and gastrulation stages in species with equal/conditional development to earlier cleavage stages in species with unequal/autonomous spiral cleavage.

## Shared expression domains of orthologous TFs in annelid gastrulae

To start to investigate the implications of temporal shifts in TF activation during spiral cleavage in *O. fusiformis* and *C. teleta*, we focused on seven TFs with well-known roles during animal development (*pax2/5/8*, *tbx2/3*, *vsx2*, *AP2*, *uncx*, *HNF4* and *prop1*) (Achim et al, 2018; Angotzi et al, 2011; Arenas-Mena, 2013; Denes et al, 2007; Erclik et al, 2008; Focareta et al, 2014; He et al, 2023; Lacin et al, 2020; Martin-Duran and Hejnol, 2015; Miller et al, 1992; Planques et al, 2019; Seudre et al, 2022a; Stolfi and Levine, 2011; Tian et al, 2021; Valencia et al, 2021; Vellutini and Hejnol, 2016; Wollesen et al, 2015; Xing et al, 2021) that follow the most common temporal change between these two annelids (Fig. 5D,F), namely expressed at or after organiser specification in *O. fusiformis* (5 hpf and gastrula) but at earlier time points in *C. teleta*. In agreement with the detected expression in the RNA-seq time course (Dataset EV39), none of these genes show expression during cleavage until 5 hpf in *O. fusiformis* (Fig. 6A–G). The only exception is *pax2/5/8*, which, as expected, exhibits broad maternal expression throughout the embryo (Fig. 6A). At 5 hpf, however, all TFs show restricted expression domains (Fig. 6A–G). *pax2/5/8* is expressed in a few apical ectodermal cells that extend to encircle

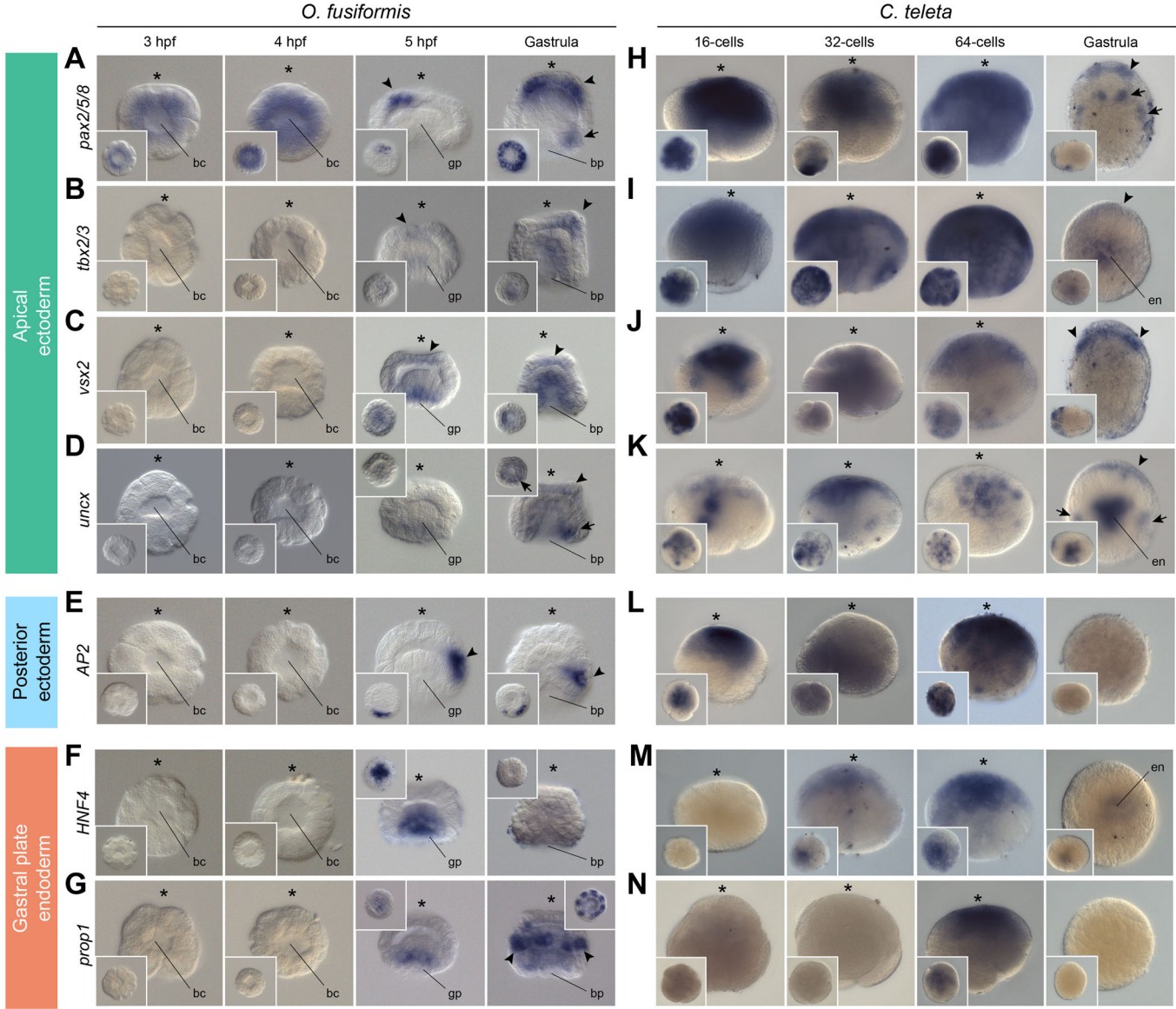

**Figure 6.   Spatial expression dynamics of orthologous transcription factors in *O. fusiformis* and *C. teleta*.**

(A–N) Whole-mount in situ hybridisation of seven orthologous transcription factors (*pax2/5/8*, *tbx2/3*, *vsx2*, *uncx*, *AP2*, *HNF4*, and *prop1*) during mid and late cleavage and at the gastrula stage in *O. fusiformis* (A–G) and *C. teleta* (H–N). In *O. fusiformis*, *pax2/5/8* (A), *tbx2/3* (B), *vsx2* (C) and *uncx* (D) are expressed in the anterior neuroectoderm (arrowheads), as well as in the posterior blastoporal rim (*pax2/5/8* and *uncx*, arrows) and endoderm (*tbx2/3* and *vsx2*). *AP2* (E) is expressed in the posterior ectoderm (arrowhead), and *HNF4* (F) and *prop1* (G) are in the gastral plate before gastrulation and in seven equatorial clusters of two ectodermal cells in the gastrula (*prop1*, arrowheads). No expression is detected before 5 hpf for any of these genes. In *C. teleta*, *pax2/5/8* (H), *tbx2/3* (I), *vsx2* (J), *uncx* (K) and *AP2* (L) are broadly expressed during mid and late cleavage. In the gastrula, *pax2/5/8*, *tbx2/3*, *vsx2* and *uncx* are detected in the anterior neuroectoderm (arrowheads), four cell clusters posterior to the foregut (*pax2/5/8*, arrows), the endoderm (*tbx2/3* and *uncx*) and two lateral mesodermal clusters (*uncx*, arrows). *HNF4* (M) is broadly detected at 32- and 64-cells and in the endoderm of the gastrula, and *prop1* (N) expression is only apparent at the 64-cell stage. Insets are animal/vegetal views, except for the gastrula stage in (H–N), which are lateral views. All images come from representative specimens of at least two biological replicates. Asterisks indicate the animal pole. bc blastocoele, bp blastopore, en endoderm, gp gastral plate.

the entire anterior apical ectoderm in the gastrula (Fig. 6A). Additionally, *pax2/5/8* is detected on one side, potentially the posterior, of the blastopore in *O. fusiformis* (Fig. 6A). As with *pax2/5/8*, *tbx2/3*, *vsx2*, and *uncx* are also detected in the apical ectoderm at 5 hpf and gastrula stages in *O. fusiformis*, but also in the gastral plate, endoderm, and the posterior blastopore rim for *uncx* (Fig. 6B–D). The TF *AP2* shows expression in discrete

ectodermal cells in the future dorsoposterior side of the embryo (Fig. 6E). Finally, *HNF4* and *prop1* are expressed in the gastral plate at 5 hpf (Fig. 6F,G). While we did not detect expression of *HNF4* at the gastrula stage for *O. fusiformis*, *prop1* is expressed in the internalised cells and seven further bilaterally symmetrical clusters of two cells around the equatorial ectoderm of the gastrula (Fig. 6F,G).

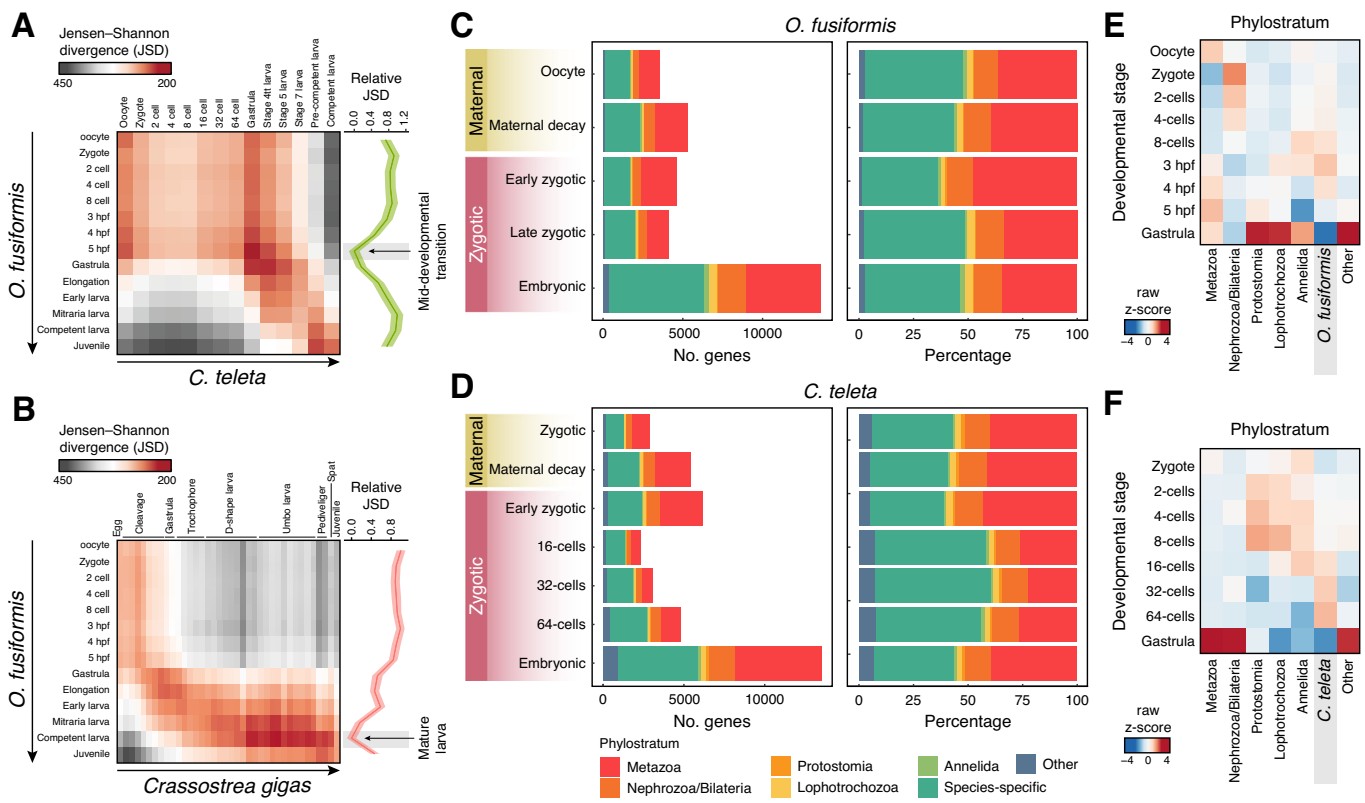

**Figure 7. A mid-developmental transition in spiral cleavage.**

(A, B) Jensen-Shannon transcriptomic divergence between all possible inter-species pairwise comparisons during the entire life cycle, from oocyte to juvenile or competent larva of *O. fusiformis* and *C. teleta* (A) and *O. fusiformis* and *C. gigas* (B). The point of maximal transcriptomic similarity between these annelids occurs at the late cleavage and gastrulation, while it happens at the ciliated larval stage between *O. fusiformis* and *C. gigas*. (C, D) Bar plots indicate the distribution of expressed genes according to their age or phylostratum at each cluster of temporally coregulated genes in *O. fusiformis* (C) and *C. teleta* (D). More ancestral genes comprise around half of the transcriptome during spiral cleavage in both species, except in the stage-specific clusters of *C. teleta* at the 16-cell, 32-cell, and 64-cell stages. (E, F) The heatmaps display the stages with the highest gene expression, categorised by age. In *O. fusiformis* (E), metazoan, protostomian, and lophotrochozoan genes are more expressed at the 5 hpf and gastrula, while in *C. teleta* (F), metazoan and bilaterian genes are highly expressed at the gastrula stage.

Consistent with in silico datasets (Dataset EV39), these seven TFs were expressed during mid and late cleavage in *C. teleta* (Fig. 6H–N). However, unlike in *O. fusiformis*, the early expression of these TFs in *C. teleta* is broad and generally distributed throughout all embryonic blastomeres for most of the genes, and more restricted expression domains only appeared at the gastrula stage (Fig. 6H–N). The early broad expression was unexpected and warrants further investigation, for example, through gene-specific knockouts (Neal et al, 2019). At the gastrula stage, *pax2/5/8* is detected in the anterior ectoderm (i.e. the developing brain) and four ventral clusters posterior to the foregut (Fig. 6H). Similarly, *tbx2/3*, *vsx2* and *uncx* are also expressed in the anterior neuroectoderm of the gastrula. Additionally, *tbx2/3* and *uncx* show expression in the internalised endoderm and *uncx* is also detected in two lateral domains, reminiscent of the most lateral clusters of *pax2/5/8* that might correspond to the developing mesoderm (Fig. 6I,J). Finally, we did not observe expression for *AP2* and *prop1* in the gastrula, but *HNF4* is weakly expressed in the endoderm of *C. teleta* at that stage (Fig. 6L–N). Altogether, the comparison of the expression domains of these seven TFs is consistent with the higher transcriptional similarity at the gastrula stage between *O. fusiformis* and *C. teleta* (Fig. 4A), as most of these genes (*pax2/5/8*, *tbx2/3*,

*vsx2*, *uncx*, and *HNF4*) have shared expression patterns in the late blastula and gastrula despite their different transcriptional dynamics during early cleavage.

## An hourglass-like dynamic of gene expression in spiral cleavage

Our findings support the gastrula as a generally conserved developmental stage––in transcriptional levels, orthogroup and gene deployment, and gene expression patterns––during early annelid embryogenesis despite the diverse morphologies at that embryonic period. Indeed, the comparative analysis of transcriptomic similarity throughout the entire development of *O. fusiformis* and *C. teleta* (from oocyte to competent larvae and juvenile stages) strengthens that the late cleavage and gastrulation stages are the most transcriptionally similar stages in the life cycle of these species, followed by the juvenile stages (Fig. 7A). The late cleavage and gastrulation thus act as a mid-developmental transition between two phases of higher transcriptomic dissimilarity, namely an earlier period during spiral cleavage and a later phase during larval development. Notably, when comparing annelids with *Crassostrea gigas*, a bivalve mollusc with autonomous spiral

cleavage, the late spiral cleavage and gastrulation are not periods of maximal transcriptomic similarity, but spiral cleavage is still a phase of high transcriptional dissimilarity (Figs. 7B and EV5). Instead, the larval stages act as stages of maximal transcriptomic similarity, consistent with previous comparisons among metazoan larvae (Martin-Zamora et al, 2023).

These intra- and inter-phyletic transcriptomic similarities remain when extending the analyses to other annelid and molluscan species (Fig. EV5). We first compared the developmental transcriptomic dynamics of *C. gigas* with those of the Mediterranean mussel *Mytilus galloprovincialis* and the abalone *Haliotis discus hannai* (Fig. EV5A,B). As observed in annelids (Fig. 7A), molluscan development is more similar at late gastrula and early trochophore stages. However, annelid and molluscan development exhibit the maximal transcriptomic similarity at the larval stages, even when including in the analysis the echiuran worm *U. unicinctus*, and with the only exception of the *O. fusiformis* and *H. discus hannai* comparison (Fig EV5C–F). Thus, despite technical and sampling differences (mainly affecting the abalone dataset), these comparisons support that the gastrula and larval stages are more transcriptomically conserved developmental stages at the phylum and spiralian level, respectively.

Deployment and enrichment in genes of ancestral origin have been associated with developmental stages of high inter-species conservation (Domazet-Loso and Tautz, 2010; Xu et al, 2016). To assess this in annelids, we assigned genes based on their phylogenetic origin and evaluated their contribution to the clusters of coregulated genes. In both species, old genes (i.e. ancestral to Metazoa and originated in Bilateria, Protostomia and Spiralia) account for more than 50% of the genes in all but the oocyte-specific clusters in *O. fusiformis* and all but the cleavage-specific clusters (16-cell, 32-cell, and 64-cell) in *C. teleta* (Fig. 7C,D). These cleavage-specific clusters exhibit more than 50% of *C. teleta*-specific genes (Fig. 7D), supporting the lack of functional annotation for many of the genes expressed at those time points in this annelid. Consistent with the gene distribution by phylostratum, the expression of genes of Metazoa, Protostomia, and Spiralia origin is higher at the gastrula stage in *O. fusiformis* (Fig. 7E), and Metazoa/Bilateria genes are highly expressed in the gastrula of *C. teleta* (Fig. 7F). However, the expression of species-specific genes is higher during mid-cleavage stages in both annelids, from the 8-cell stage to 4 hpf in *O. fusiformis* and from the 16-cell to the 64-cell stages in *C. teleta* (Fig. 7E,F). Therefore, our findings indicate that annelids undergo a mid-developmental transition in their life cycle around gastrulation when embryos deploy more ancestral genes and exhibit a high degree of transcriptomic and gene expression pattern similarity.

# Discussion

Our work profiles, with an unprecedented temporal resolution for spiral cleavage (Chou et al, 2016; Harry and Zakas, 2024; Martin-Duran et al, 2021; Zhang et al, 2012), the transcriptomic dynamics during early embryogenesis in *O. fusiformis* and *C. teleta*, two annelids with equal/conditional and unequal/autonomous spiral cleavage, respectively. Although the evolutionary distance between *O. fusiformis* and *C. teleta* likely influences some of our observations, our extensive dataset defines a novel comparative

framework to investigate the genetics and mechanisms controlling and diversifying spiral cleavage, one of the most ancient and broadly conserved cleavage programmes (Hejnol, 2010; Henry, 2014; Martin-Duran and Marletaz, 2020).

## The early transcriptomic dynamics in spiral cleavage

The transition from a maternally regulated development to one controlled by the zygotic genome––the so-called maternal-to-zygotic transition––is a fundamental event in early animal embryogenesis (Brantley and Di Talia, 2024; Lee et al, 2014; Tadros and Lipshitz, 2009; Vastenhouw et al, 2019). This involves the degradation of maternal transcripts and proteins, as well as the epigenetic remodelling of the zygotic genome to enable transcription (i.e. zygotic genome activation). These often, but not always, coincide with the desynchronisation of cell cycles between different embryonic regions (Lu et al, 2009; Newport and Kirschner, 1982; O'Farrell et al, 2004). Notably, the timing of these events varies dramatically, even between phylogenetically closely related lineages (O'Farrell et al, 2004; Vastenhouw et al, 2019). Yet, how and when these critical steps occur during embryogenesis remains unknown in most animals, especially those exhibiting spiral cleavage. In the annelid leech *Helobdella triserialis*, which displays a modified spiral cleavage (Dohle, 1999), embryonic transcription begins early, when the embryo has ~20 cells, only in a subset of blastomeres, and is essential to control spindle orientation (Bissen and Smith, 1996). Based on transcriptomic profiling, zygotic genome activation appears to occur around the 64-cell stage in the annelid *P. dumerilii* (Chou et al, 2016). However, only two earlier time points (the 8- and 30-cell stages) were sampled in that study. In *O. fusiformis* and *C. teleta*, maternal transcripts are largely cleared in two waves by the 16- and 32-cell stages, and the first signs of gene expression occur around the 4- and 8-cell stages, with the differential upregulation of genes at the 16-cell stage (the fourth cell cycle) being essential for normal development in both species (Figs. 2D–F, 3A,B, 4B,C, 8A). This is consistent with a recent report on the zygotic genome activation of the annelid with equal/conditional spiral cleavage *Ophelia limacina* occurring at the 8-cell stage (Grinberg et al, 2025). Notably, the first cleavage asymmetry in *O. fusiformis* occurs two cell cycles later (64-cell stage or 5 hpf), with the formation of the 4q micromeres, one of which will become the embryonic organiser (Seudre et al, 2022a). Therefore, the maternal-to-zygotic transition may be a very early phenomenon (occurring between the 4- and 16-cell stages) and not coincide with the desynchronisation of cell divisions in annelids; yet inter-species variation is likely. A more extensive taxon sampling that includes other groups with spiral cleavage, along with more precise approaches to measuring zygotic transcription and maternal transcript dynamics than poly(A)-selection based RNA-seq (Lee et al, 2014; Vastenhouw et al, 2019), will provide a more comprehensive view of the maternal-to-zygotic transition in Spiralia.

Although it happens at similar developmental time points, the zygotic genome activation appears to be more pronounced in *C. teleta* than in *O. fusiformis* at the 16-cell stage (Figs. 2E,F and 8A). This coincides and is consistent with the specification of the embryonic organiser and the activation of head patterning programmes at this stage in *C. teleta* (Fig. 1D) (Amiel et al, 2013; Carrillo-Baltodano and Meyer, 2017). Similarly, the specification of the embryonic organiser by the 64-cell stage is concomitant with an increase in gene upregulation in *O. fusiformis* (Fig. 2E) (Seudre et al, 2022a). As observed in the leech *H. triserialis* (Bissen and Smith, 1996), our data also support that zygotic genome activation

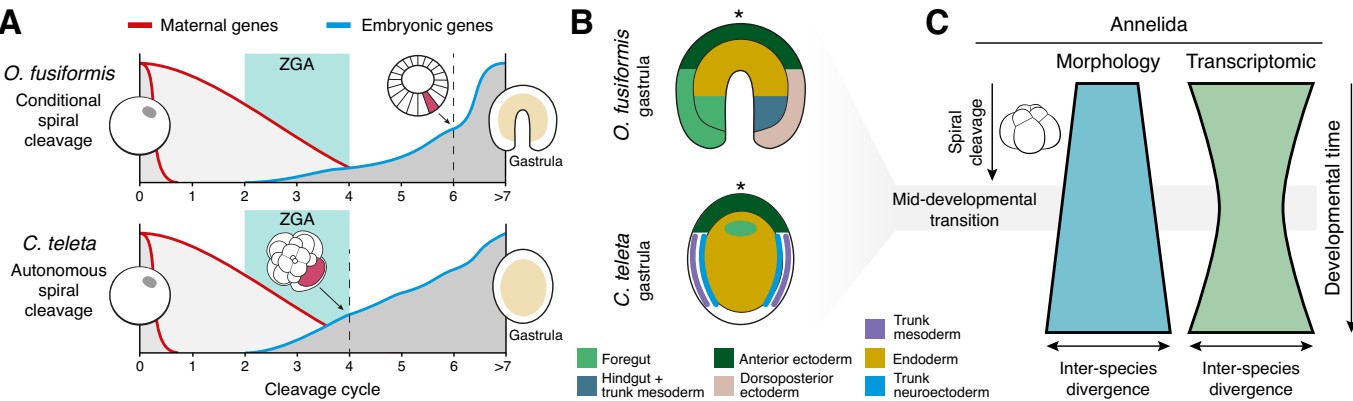

**Figure 8. Transcriptional and morphological dynamics are decoupled in spiral cleavage.**

(A) Schematic summary of the transcriptomic dynamics during spiral cleavage in *O. fusiformis* and *C. teleta*. Two waves of mRNA decay (one that is oocyte/zygote-specific and another that extends to about the 8-cell stage) occur during early cleavage. Zygotic genome activation and the maternal-to-zygotic transition (MZT) likely occur between the second (4-cell stage) and fourth cell division (16-cell stage) and more intensely in *C. teleta*, as it coincides with the specification of the embryonic organiser (schematic drawing and dotted vertical line). The large increase in zygotic transcriptional activity occurs at the sixth cell division (64-cell stage) in *O. fusiformis*, when the embryonic organiser is established in this species (dotted line and schematic drawing). (B) Schematic comparison of broad areas of developmental competence at the gastrula stage between *O. fusiformis* and *C. teleta* (see main text for details). (C) Differently from other animal phyla, morphological and transcriptomic similarities are uncoupled during development in Annelida. The early embryonic phase of spiral cleavage is morphologically stereotypical but transcriptionally divergent. However, annelid embryos converge into a broad transcriptomic and molecular patterning similarity phase by the gastrula stage, which acts as a mid-developmental transitional period. Annelid embryos diverge (morphologically and transcriptionally) upon gastrulation as they develop into lineage-specific larval forms. Drawings are not to scale.

is not uniform throughout the embryo, at least in *O. fusiformis*, as some of the upregulated genes, such as the expanded *foxQ2* genes (Seudre et al, 2022b) and *fjx1*, exhibit asymmetric expression along the animal-vegetal axis at these early stages of development. These markers concentrate in the animal ectodermal pole (Seudre et al, 2022b) (Fig. 3E), which generates anterior neuroectodermal derivatives in most spiralians and bilaterians (Martin-Duran and Hejnol, 2021; Nielsen, 2004, 2005). This reinforces the hypothesis that anterior neural structures develop early and more autonomously in embryos with both autonomous and conditional modes of spiral cleavage (Carrillo-Baltodano and Meyer, 2017). In equal/conditional spiral cleavage, the posterior/vegetal pole likely remains uncommitted as new tiers of micromeres are produced (Morrill et al, 1973; van den Biggelaar and Guerrier, 1979) and until the inductive signalling event mediated by the FGF-ERK1/2 pathway primes one of the blastomeres to become the embryonic organiser (Seudre et al, 2022a; Tan et al, 2022, 2023), restricting anterior fates and activating posterodorsal developmental on one side of the embryo (Seudre et al, 2022a). Therefore, although embryos with equal/conditional and unequal/autonomous spiral cleavage share general transcriptional dynamics during early embryogenesis (Fig. 8A), the different modes of cell-fate specification, which determine the timing of axial patterning and the onset of body regionalisation, outweigh the conservation of the cleavage programme and cell lineages in shaping specific transcriptional trends during early embryogenesis.

## The transcriptomic convergence during axial patterning in annelid embryos

Despite their different early transcriptional dynamics and modes of cell fate specification, late cleavage and gastrula stages are the most similar between *O. fusiformis* and *C. teleta*. The analysis of seven

transcription factors activated late (5 hpf and gastrula) in *O. fusiformis* but much earlier in *C. teleta* demonstrates that similarity in transcription levels at the gastrula stage is often accompanied by similarity in expression domains (Fig. 6). This is consistent with studies in these annelids of candidate transcription factors with evolutionarily conserved expression domains across distantly related animals, which act as "marker" genes for the development of specific structures. Foregut genes, such as *nkx2.1*, *foxA* and *gsc*, are expressed in the prospective oral ectoderm of the gastrula, while endodermal markers, such as *GATA4/5/6* paralogs, localise to the internalised cells (Fig. 8B) (Boyle and Seaver, 2008; Boyle et al, 2014; Martin-Duran et al, 2016). Likewise, anterior neuroectodermal genes (e.g. *sox* genes and those in this study) are expressed in the animal/anterior pole in the gastrulae of *O. fusiformis* and *C. teleta* (Fig. 8B) (Carrillo-Baltodano et al, 2024; Meyer and Seaver, 2009; Sur et al, 2017). Notably, however, the two species differ in the extent and levels of trunk mesodermal and neuroectodermal gene expression (e.g. *pax3/7*, *evx*, *cdx*, *twist*). While the precursors of the ecto- and mesodermal trunk occupy a large lateral domain in the gastrula of *C. teleta* (Dill et al, 2007; Frobius and Seaver, 2006; Seaver et al, 2012), these are restricted to one side of the blastoporal rim and the few possible descendants of the embryonic organiser in *O. fusiformis* (Martin-Duran et al, 2018; Martin-Duran et al, 2016; Seudre et al, 2022a) (Fig. 8B). These differences reflect the temporal and developmental differences in trunk formation between these two species, which are ultimately linked to their distinct larval types and life cycle strategies (Martin-Zamora et al, 2023). Equally, in annelids with heteromorphic larval types (i.e. planktotrophic and lecithotrophic), such as *Streblospio benedicti*, genes exhibiting temporal shifts are connected to morph-specific developmental and physiological traits, like the formation of a functional gut in the planktotrophic type and the metabolism of the maternal nourishment in the lecithotrophic type (Harry and Zakas, 2024). Therefore,

despite the conservation of the cleavage programme and the early transcriptional differences, a shared body and molecular patterning that reflects subsequent species-specific developmental traits only emerge during gastrulation in annelid embryogenesis.

## A mid-developmental transition in annelid development

In some animal clades, there is a period during mid-development when phylogenetically related taxa exhibit a significant degree of morphological and transcriptomic similarity (Kalinka and Tomancak, 2012; Slack et al, 1993; Waddington, 1956). This is thought to be the stage at which the body plan's essential traits of the clade are established; thus, this phase is also termed the phylotypic stage (Duboule, 1994; Sander, 1994). Although the phylotypic stage concept is contested (Bininda-Emonds et al, 2003; Hall, 1997; Hejnol and Dunn, 2016; Richardson, 1995), an equivalent phenomenon has been described during the development of plants, fungi, and brown algae (Cheng et al, 2015; Lotharukpong et al, 2024; Quint et al, 2012). Therefore, the phylotypic stage may reflect an emergent property of multicellular systems that occurs when genetic programmes become active and interact to define the different body regions, creating a stage that is more sensitive to evolutionary change (Bogdanovic et al, 2016; Galis and Metz, 2001; Liu et al, 2021; Marletaz et al, 2018; Tena et al, 2014; Uchida et al, 2022; Uesaka et al, 2019; Zalts and Yanai, 2017). From a morphological standpoint, however, the highest degree of embryonic similarity amongst the disparate and distantly related animal groups with spiral cleavage occurs during the earliest zygotic divisions (Fig. 8C). This has led some authors to claim that there is no phylotypic stage in Spiralia (Wu et al, 2019), while others have argued that the larva is the phylotypic stage in annelids and molluscs (Cohen and Massey, 2008; Levin et al, 2016; Paps et al, 2015; Xu et al, 2016). Yet, life cycles are vastly diverse in Spiralia, and larvae differ dramatically in their morphology and ecology, even within a phylum (Liang et al, 2024; Rouse, 2008). Thus, a mid-developmental phylotypic stage in Spiralia has remained a matter of controversy.

Our data highlights the gastrula as a critical developmental time point in the spiralian embryos we studied (Fig. 8B,C). Although the amount of maternally provided nourishment heavily influences embryonic morphologies and morphogenetic processes at this stage (e.g. gastrulation by invagination or epiboly), the gastrulae of *O. fusiformis* and *C. teleta* exhibit high overall transcriptomic similarity, shared molecular expression patterns, and areas of developmental competence (Fig. 8B; see above). Morphological and transcriptomic similarities are thus uncoupled during spiral cleavage, indicating that the extreme conservation of this early embryogenesis relies on a limited gene set and/or maternal proteins contributing to conserved cellular physical mechanics (Brun-Usan et al, 2017). Moreover, developmental system drift in the specification of homologous cell types and body regions may be widespread in these lineages, even between closely related species (Carrillo-Baltodano et al, 2025; Lanza and Seaver, 2020). Therefore, unlike in other animal groups, such as vertebrates and insects, a mid-developmental phylotypic stage occurring around gastrulation is only evident at the transcriptomic level in Spiralia. This stage marks the transition from an early morphologically similar but transcriptionally divergent phase of cleavage to a late morphologically and transcriptionally divergent period during organogenesis that results in the formation of a larva or juvenile (Fig. 8C). This scenario challenges previous interpretations of spiralian

development (Cohen and Massey, 2008; Levin et al, 2016; Paps et al, 2015; Wu et al, 2019; Xu et al, 2016), proposing a stage in embryogenesis where the astonishing body plan diversity of animal lineages with spiral cleavage may emerge.

In summary, our work provides an unparalleled characterisation of the transcriptional dynamics during early embryogenesis in *O. fusiformis* and *C. teleta*. These two annelids exhibit spiral cleavage but autonomous and conditional specification of their body plans, respectively. Unexpectedly, the transcriptomic profiles during their early development are highly divergent, influenced by their strategies and different timings to define the embryonic organiser and bilateral symmetry. This indicates that the high conservation of the spiralian cleavage programme does not constrain the deployment of transcriptional programmes during the development of these annelids. Thus, maternal proteomes and/or biophysical properties might play a more significant role in maintaining the spiral cleavage pattern of cell divisions. Yet, the embryos of these two annelids converge in their transcriptomic and body patterning profiles at gastrulation, which appears as a mid-developmental transitional phase in annelid embryogenesis. In the future, exploiting low-input epigenomic profile approaches, such as ATAC-seq and CUT&Tag (Buenrostro et al, 2015; Kaya-Okur et al, 2019), will complement our transcriptomic resources and help infer the functional and regulatory dynamics underpinning the differences in gene expression between species with autonomous and conditional spiral cleavage. Altogether, our study provides a unique, high-resolution resource for investigating the early stages of animal embryogenesis in two emerging spiralian models, proposing new conceptual scenarios to explore the developmental mechanisms that control spiral cleavage, an iconic and ancient mode of animal development.

## Methods

**Reagents and tools table**

| Reagent/resource | Reference or source | Identifier or catalogue number |
| --- | --- | --- |
| **Experimental models** | | |
| *Owenia fusiformis* | Field collection | |
| *Capitella teleta* | In-house culture | |
| **Recombinant DNA** | | |
| **Antibodies** | | |
| Mouse anti-acetylated α-tubulin (clone 6-11B-1) | Merck-Sigma | #MABT868 |
| mouse anti-beta-tubulin | Developmental Studies Hybridoma Bank | E7 |
| mouse anti-RNA polymerase II, clone CTD4H8 | Merck-Sigma | #05623 |
| Rabbit anti-serotonin | Merck-Sigma | #S5545 |
| AlexaFluor-conjugated secondary antibodies | ThermoFisher Scientific | A-21428, A-32731, A-21235 |
| **Oligonucleotides and other sequence-based reagents** | | |
| PCR primers | This study | Appendix Table S4 |
| Q5® High-Fidelity 2X Master Mix | New England Biolabs | M0492L |

| Reagent/resource | Reference or source | Identifier or catalogue number |
|---|---|---|
| **Chemicals, Enzymes and other reagents** | | |
| Monarch Total RNA Miniprep kit | New England Biolabs | T2010S |
| RNA ScreenTape | Agilent | 5067-5576 |
| RNA ScreenTape Ladder | Agilent | 5067-5578 |
| RNA ScreenTape Sample Buffer | Agilent | 5067-5577 |
| Qubit™ RNA High Sensitivity (HS) | Thermo Fisher Scientific | Q32852 |
| Qubit™ RNA High Sensitivity (BR) | Thermo Fisher Scientific | Q10210 |
| MEGAscript™ T7 Transcription Kit | Thermo Fisher Scientific | AM1334 |
| Actinomycin D | Cell Signaling Technology | #15021 |
| AlexaFluor 488 Phalloidin | Thermo Fisher Scientific | A12379 |
| **Software** | | |
| Kallisto (0.46.2) | Bray et al, 2016 | |
| DESeq2 (1.30.1) | Love et al, 2014 | |
| Limma (3.46.0) | Ritchie et al, 2015 | |
| EnhancedVolcano (1.8.0) | Blighe et al, 2018 | |
| Mfuzz (2.50.0) | Kumar and M, 2007 | |
| NbClust (3.0.1) | Charrad et al, 2014 | |
| ComplexHeatmap (2.6.2) | Gu, 2022; Gu et al, 2016 | |
| OrthoFinder (2.5.5) | Emms and Kelly, 2019 | |
| TopGO (2.42.0) | Alexa and Rahnenfuhrer, 2024 | |
| Scikit-learn (1.7) | Pedregosa et al, 2011 | |
| KEGGREST (1.30.1) | Tenenbaum and Maintainer, 2024 | |
| Pheatmap (1.0.13) | Kolde R (2025) | |
| ClusterProfiler (3.18.1) | Wu et al, 2021 | |
| Biostrings (2.30.1) | Pagès et al, 2024 | |
| seqinR (4.2-30) | Charif and Lobry, 2007 | |
| coRdon (1.24.0) | Elek et al, 2024 | |
| Primer3 (4.1.0) | Untergasser et al, 2012 | |
| **Other** | | |
| Illumina NovaSeq6000 | Illumina | |
| Leica DMRA2 microscope | Leica | |
| Leica Stellaris 8 confocal microscope | Leica | |

## Methods and protocols

### Animal culture and sample collection

Sexually mature *Owenia fusiformis* Delle Chiaje, 1844 adults were collected from subtidal waters near the Station Biologique de Roscoff and cultured in the laboratory as previously described (Carrillo-Baltodano et al, 2021). In vitro fertilisation and collection

of embryonic stages were performed as previously outlined (Carrillo-Baltodano et al, 2021). *Capitella teleta* (Blake, Grassle and Eckelbarger, 2009) was cultured, and embryos were collected following established protocols (Seaver, 2005). For oocyte collection, females whose offspring were used to collect cleavage stages were isolated and dissected after 1 week with abundant mud to obtain their oocytes.

### Library preparation and sequencing

*O. fusiformis* developmental samples encompassing active oocyte, zygote, 2-cell, 4-cell and 8-cell stages, 3 h post-fertilisation (hpf), 4 and 5 hpf (blastula), and gastrula (9 hpf) were collected in duplicates, flash frozen in liquid nitrogen, and stored at −80 °C for total RNA extraction. Samples within replicates were paired, with each one containing 500 embryos coming from the same in vitro fertilisation. Reciprocal developmental stages for *C. teleta*, namely the oocyte, zygote, 2-, 4-, 8-, 16-, 32-, and 64-cell stages, and gastrulae were collected in duplicates using similar genetic pools of different brood tubes. Total RNA was isolated using a Monarch Total RNA Miniprep kit (New England Biolabs), RNA quality was assessed using the Agilent RNA TapeStation (Agilent), and RNA quantity was measured with the Qubit RNA HS Assay Kit (Thermo Fisher Scientific) before library preparation. Following the supplier's recommendations, we used to prepare strand-specific mRNA Illumina libraries that were sequenced at the Oxford Genomics Centre (University of Oxford, UK) over three lanes of an Illumina NovaSeq6000 system in $2 \times 150$ bp mode to a depth of around 50 million reads per sample (Appendix Tables S2 and S3).

### RNA-seq analyses

To profile gene expression dynamics during development, we first utilised Kallisto (0.46.2) (Bray et al, 2016) to map RNA-seq reads to the reference gene models of *O. fusiformis* and *C. teleta*. Mapping statistics are provided in Appendix Tables S2 and S3. Quality control was conducted using TPM (transcripts per million) matrices, visualising read distributions, ridge plots, and scatter plots of sample correlations. Genes with TPM >2 were defined as expressed in each developmental stage. Gene expression across all samples was normalised using DESeq2 (Love et al, 2014), and correlation coefficients were calculated to elucidate relationships among developmental stages. For *O. fusiformis*, batch effects between the two replicates were removed using the limma package (Ritchie et al, 2015). To identify differentially expressed genes (DEGs) between adjacent developmental stages, pairwise comparisons were performed using DESeq2 (Love et al, 2014), and DEGs were visualised with EnhancedVolcano (Blighe et al, 2018). To recover subtle changes and allow the inclusion of all differences in expression between stages, the log2(fold change) cutoff was set to 0.

### Gene clustering

To elucidate genes with similar expression patterns, we applied soft clustering using the Mfuzz package (Kumar and M, 2007), clustering gene expression profiles based on the average normalised values from DESeq2. To determine the optimal number of clusters, we applied the 'centroid' clustering method, 'Euclidean' distance with 'ch' index by the R package NbClust (version 3.0.1) (Charrad et al, 2014). Hierarchical clustering was also performed to resolve relationships among developmental stages. Clusters were categorised according to their expression profiles, particularly the

stages of highest expression, along with insights from hierarchical clustering and PCA analyses. Gene expression dynamics across the whole profile and within individual clusters were visualised using ComplexHeatmap (Gu, 2022; Gu et al, 2016) and line plots. Furthermore, each group was characterised for transcription factor composition based on TF families, and the results were visualised numerically and proportionally using Nightingale rose charts.

### Jensen-Shannon distance (JSD) comparisons

We employed the Jensen-Shannon distance (JSD) (Endres and Schindelin, 2003; Lin, 1991) to estimate inter-sample distances and explore sample relationships across developmental stages between species. For post-gastrula stages in *O. fusiformis*, *C. teleta* and other annelid and molluscan species, we used publicly available datasets (Dataset EV40) (Burns and Pechenik, 2017; Data ref: Burns and Pechenik, 2017; Chou et al, 2016; Data ref: Chou et al, 2016; Harry and Zakas, 2024; Data ref: Harry and Zakas, 2024; Martin-Zamora et al, 2023; Data ref: Martin-Zamora et al, 2023; Miglioli et al, 2024; Data ref: Miglioli et al, 2024; Park et al, 2018; Data ref: Park et al, 2018; Xu et al, 2016; Data ref: Xu et al, 2016). We first identified all one-to-one orthologous genes between the species with OrthoFinder (Emms and Kelly, 2019) and extracted their corresponding TPM values. These TPM values were then quantile-transformed using the Quantile-Transformer (Scikit-learn (1.7)) Python script to convert them into probability distributions (Pedregosa et al, 2011). We calculated raw JSD values for each pair of developmental stages across species using these distributions. To ensure comparability of JSD values across different stages, we normalised the raw JSD values using a method described in (Martin-Zamora et al, 2023). The raw JSD values were visualised in a heatmap, while the normalised values were presented in a line plot alongside the raw data.

### Heterochrony analyses between species

To define gene expression thresholds for downstream heterochrony analysis, we examined the transcriptome-wide distributions of $\log_2$-transformed TPM values from publicly available datasets for each species. These distributions exhibited asymmetric unimodal patterns, and expression cutoffs were defined based on dataset-specific inflexion points: *O. fusiformis* and *C. teleta*: TPM $\geq 2$ ($\log_2$TPM $\geq 1$); *Platynereis dumerilii*: TPM $\geq 1$ ($\log_2$TPM $\geq 0$); *Crassostrea gigas*: TPM $\geq 4$ ($\log_2$TPM $\geq 2$); *Urechis unicinctus*: TPM $\geq 16$ ($\log_2$TPM $\geq 4$). Heterochronic shifts were identified using one-to-one orthologous genes across species, with a focus on early developmental stages. The equivalent early developmental stages were excluded from specific pairwise comparisons. In *U. unicinctus* vs. *O. fusiformis* and *U. unicinctus* vs. *C. teleta* comparisons, the following stages were excluded: 1-cell, 2-cell, 4-cell, 8-cell and blastula. In the *C. gigas* vs. *O. fusiformis* and *C. gigas* vs. *C. teleta* comparisons, 1-cell, 2-cell, 4-cell, 8-cell, blastula and gastrula were excluded.

### Inter-species inter-clusters comparison analysis

To investigate the global similarity between the two species beyond one-to-one ortholog comparisons, we performed inter-cluster comparisons based on the complete set of orthologous relationships, including one-to-one, one-to-many, and many-to-many orthologues, as defined by OrthoFinder (Emms and Kelly, 2019). For each cluster pair across the two species, we conducted a hypergeometric test to evaluate the statistical significance of gene overlap, and the *p* values were adjusted using the Benjamini–Hochberg method. The adjusted *p*

values were visualised as a heatmap using the pheatmap package (Kolde, 2025) in RStudio, with values represented on a $-\log_{10}(p)$ transformation.

### GO and KEGG enrichment

We performed GO term enrichment analyses using the TopGO package (Alexa and Rahnenfuhrer, 2024) to explore the possible biological functions of each cluster. The GO universe was defined based on the annotations in Datasets EV41 and EV42. Enrichment statistics were calculated using the Fisher test, and the top 10 or 15 enriched GO terms for each cluster were selected for visualisation. In KEGG pathway enrichment analyses, we first retrieved pathway names using the KO IDs listed in Datasets EV43 and EV44 with the KEGGREST package (Tenenbaum and Maintainer, 2024). After constructing the KEGG annotation table, we performed enrichment analysis using the ClusterProfiler package (Wu et al, 2021), setting a *p* value cutoff of 0.05 and a *q* value cutoff of 0.5 without *p* value adjustment. The results of KEGG pathway enrichment were presented in dot plots for clarity and interpretation.

### Phylostratigraphy analysis

To investigate the contribution of evolutionary genes to development, we used available gene ages based on previous gene family evolutionary analyses (Martin-Zamora et al, 2023). The proportion of genes within each phylostratum was calculated according to their developmental origins across the entire gene profile and visualised using bar plots. To assess the abundance of these genes across developmental stages, we performed quantile normalisation for each stage, and the normalised values were presented in a heatmap to illustrate their distribution and trends.

### Codon usage

To determine whether there is a bias in codon usage among different clusters, we extracted the transcripts within each cluster and calculated their codon usage individually using the R packages biostrings, seqinR, and coRdon (Charif and Lobry, 2007; Elek et al, 2024; Pagès et al, 2024). To identify changes in codon usage, pairwise comparisons were performed for each codon, with the $\log_2$(Ratio) used to represent the magnitude of change between clusters. These changes were visualised in bar plots, where, for *O. fusiformis*, codons on the left side of the plot were identified as unstable, while those on the right were classified as stable.

### Gene cloning and in situ hybridisation

To generate riboprobes for whole-mount in situ hybridisation and gene expression analyses, the coding sequences of candidate genes were extracted from the genomes of both species, and gene-specific primers were designed via the Primer3 web tool (Untergasser et al, 2012) to amplify gene products ranging from 1000 to 1500 bp (Appendix Table S4). The genes were amplified using cDNA from a mix of developmental stages, and the amplicons were validated by Sanger sequencing. Fixation and whole-mount in situ hybridisation (ISH) were performed according to previously published protocols (Carrillo-Baltodano et al, 2021; Martin-Duran et al, 2016; Seaver, 2005; Seudre et al, 2022a).

### Actinomycin D treatments

Actinomycin D (Cell Signaling Technology, #15021) 10 mM stocks were made in dimethyl sulfoxide (DMSO). Working solutions were

prepared in ASW to the desired concentrations (75 µM for *O. fusiformis* and 25 µM for *C. teleta*). Embryos of *O. fusiformis* and *C. teleta* were treated in overlapping developmental time windows from the 2-cell stage, washed four times in ASW with 60 µg/mL penicillin and 50 µg/mL streptomycin (ASW + PS), and grown in ASW + PS until the 24 hpf mitraria larva (*O. fusiformis*) and stage 5 larva (*C. teleta*).

### Immunostaining

Fixation and antibody staining were performed as described elsewhere (Carrillo-Baltodano et al, 2021). The primary antibodies mouse anti-acetylated α-tubulin (clone 6-11B-1, Merk-Sigma, #MABT868, 1:800), rabbit anti-serotonin antibody (Merck-Sigma, #S5545, 1:300) mouse anti-beta-tubulin (E7, Developmental Studies Hybridoma Bank, 1:20) and mouse anti-RNA polymerase II (clone CTD4H8, Merk-Sigma, #05623, 1:200) were diluted in 5% normal goat serum (NGS) in phosphate-buffered saline with 0.5% Triton X-100 (PTx) and incubated overnight at 4 °C. After several washes in 1% bovine serum albumin (BSA) in PTx, samples were incubated with AlexaFluor-conjugated secondary antibodies (ThermoFisher Scientific, 1:600) plus DAPI (stock 2 mg/ml, 1:2000) and 1:100 AlexaFluor 488 Phalloidin (Thermo Fisher Scientific, cat#: A12379) diluted in 5% NGS in PTx overnight at 4 °C.

### Imaging

Representative embryos were cleared and mounted in 70% glycerol in phosphate-buffered saline. Whole-mount in situ hybridisation samples were imaged with a Leica DMRA2 upright microscope equipped with an Infinity5 camera (Lumenera) using differential interference contrast (DIC) optics. Confocal laser scanning microscopy (CLSM) images were taken with a Leica Stellaris 8. CLSM Z-stack projections were built with ImageJ2 (Rueden et al, 2017) and Nikon NIS-Elements software. DIC images were digitally stacked with Helicon Focus 7 (HeliconSoft). Brightness and contrast were edited with Adobe Photoshop CC, and figures were built with Adobe Illustrator CC (Adobe Inc.).

## Data availability

The new sequencing data generated in this project have been deposited at the Gene Expression Omnibus portal with BioProject accession number PRJNA1019281 (https://www.ncbi.nlm.nih.gov/bioproject/PRJNA1019281). The original confocal and differential interference contrast images have been deposited at the BioImage Archive with accession number S-BIAD2125 (https://www.ebi.ac.uk/biostudies/bioimages/studies/S-BIAD2125).

The source data of this paper are collected in the following database record: biostudies:S-SCDT-10_1038-S44319-025-00569-4.

## Peer review information

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

## Acknowledgements

We thank members of the Martín-Durán lab for their support and comments on this manuscript. This study employed computational resources from the high-performance computing facility Apocrita at Queen Mary University of London. BBSRC grant BB/W019698/1 supported the research reported in this paper. This work was also funded by the European Union Horizon 2020 Framework Programme (European Research Council Starting Grant agreement number 801669) and the Biotechnology and Biological Sciences Research Council (BB/Y004221/1) to JMMD. JW is supported by a China Scholarship Council doctoral fellowship (CSC NO. 202306330025), and AMCB is supported by a Biotechnology and Biological Sciences Research Council grant (BB/Y004221/1).

## Author contributions

**Yan Liang**: Conceptualisation; Formal analysis; Investigation; Visualisation; Writing—original draft; Writing—review and editing. **Jingcheng Wei**: Formal analysis; Investigation; Visualisation; Writing—original draft; Writing—review and editing. **Yue Kang**: Investigation. **Allan M Carrillo-Baltodano**: Data curation; Formal analysis; Investigation; Visualisation; Writing—original draft; Writing—review and editing. **José M Martín-Durán**: Conceptualisation; Data curation; Formal analysis; Supervision; Funding acquisition; Investigation; Visualisation; Writing—original draft; Project administration; Writing—review and editing.

Source data underlying figure panels in this paper may have individual authorship assigned. Where available, figure panel/source data authorship is listed in the following database record: biostudies:S-SCDT-10_1038-S44319-025-00569-4.

## Disclosure and competing interests statement

The authors declare no competing interests.

# Expanded View Figures

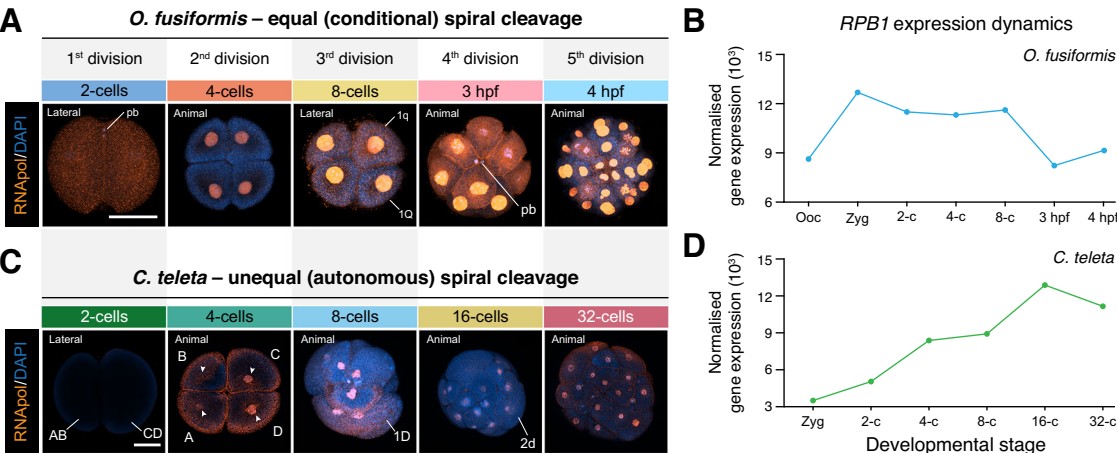

**Figure EV1. Dynamics of RNA polymerase II nuclearisation during spiral cleavage.**

(A, C) Z-projections of confocal stacks of embryos of *O. fusiformis* (A) and *C. teleta* (C) from the 2-cell stage to 4 h post-fertilisation (hpf) or the 32-cell stage. RNA polymerase II localises to the nuclei from the 4-cell stage onwards in both annelids. In *C. teleta*, the nuclearisation is more intense in the C and D blastomeres than in the A and B cells. (B, D) Expression dynamics of the *RPB1* gene (largest subunit of the RNA polymerase II, recognised by the antibody used in A and C) in *O. fusiformis* (B) and *C. teleta* (D). In the two annelids, RPB1 is a highly abundant maternal gene. Gene expression values are the average of two biological replicates.

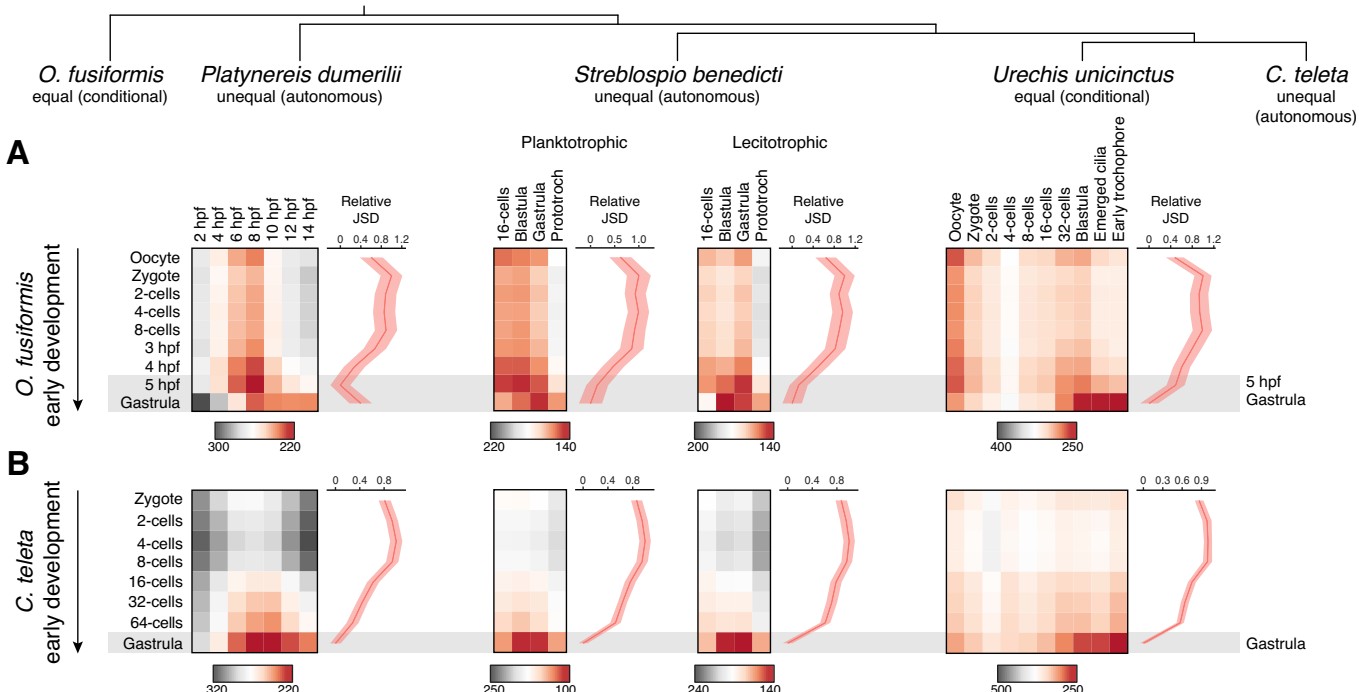

**Figure EV2. Transcriptomic dynamics during spiral cleavage in *Annelida*.**

(A, B) Jensen-Shannon transcriptomic divergence during the spiral cleavage between *O. fusiformis* (A) and *C. teleta* (B) and four other annelid species with publicly available transcriptomic resources covering at least one cleavage stage and the gastrula stage. In all cases, the point of maximal transcriptomic similarity occurs at the late cleavage and gastrulation (grey horizontal bar).

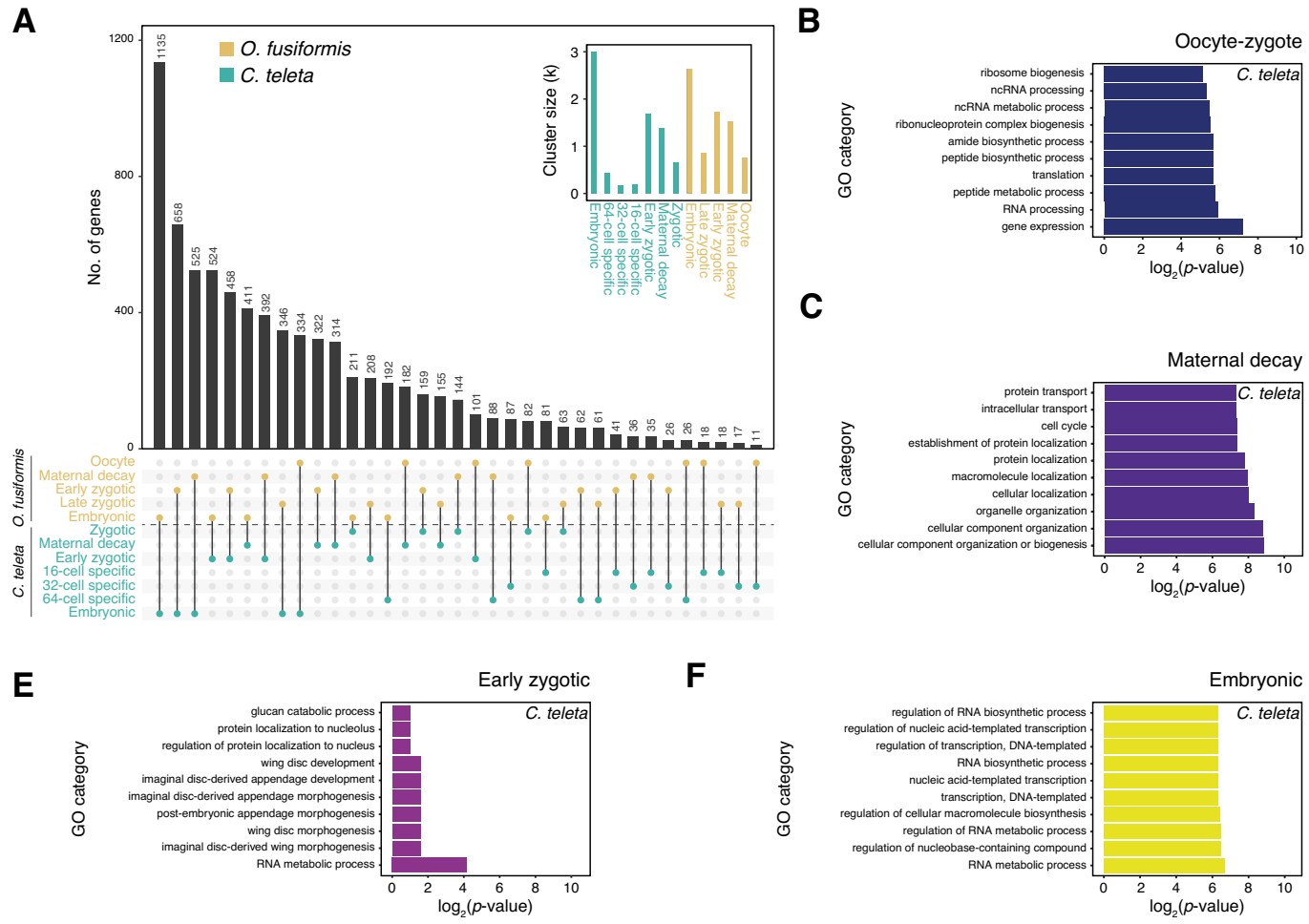

**Figure EV3. The comparison of clusters of temporally coregulated genes between conditional and autonomous spiral cleavage.**

(A) Upset plot indicating the number of shared one-to-one orthologs between clusters of temporally coregulated genes. The inset indicates the total number of genes in each cluster. (B–F) Bar plots indicating the top ten Gene Ontology (GO) categories amongst shared orthologous genes in the oocyte/zygote, maternal decay, early zygotic, and embryonic clusters, according to the GO annotation for the *C. teleta* ortholog. P values were computed from upper-tail Fisher's exact tests.

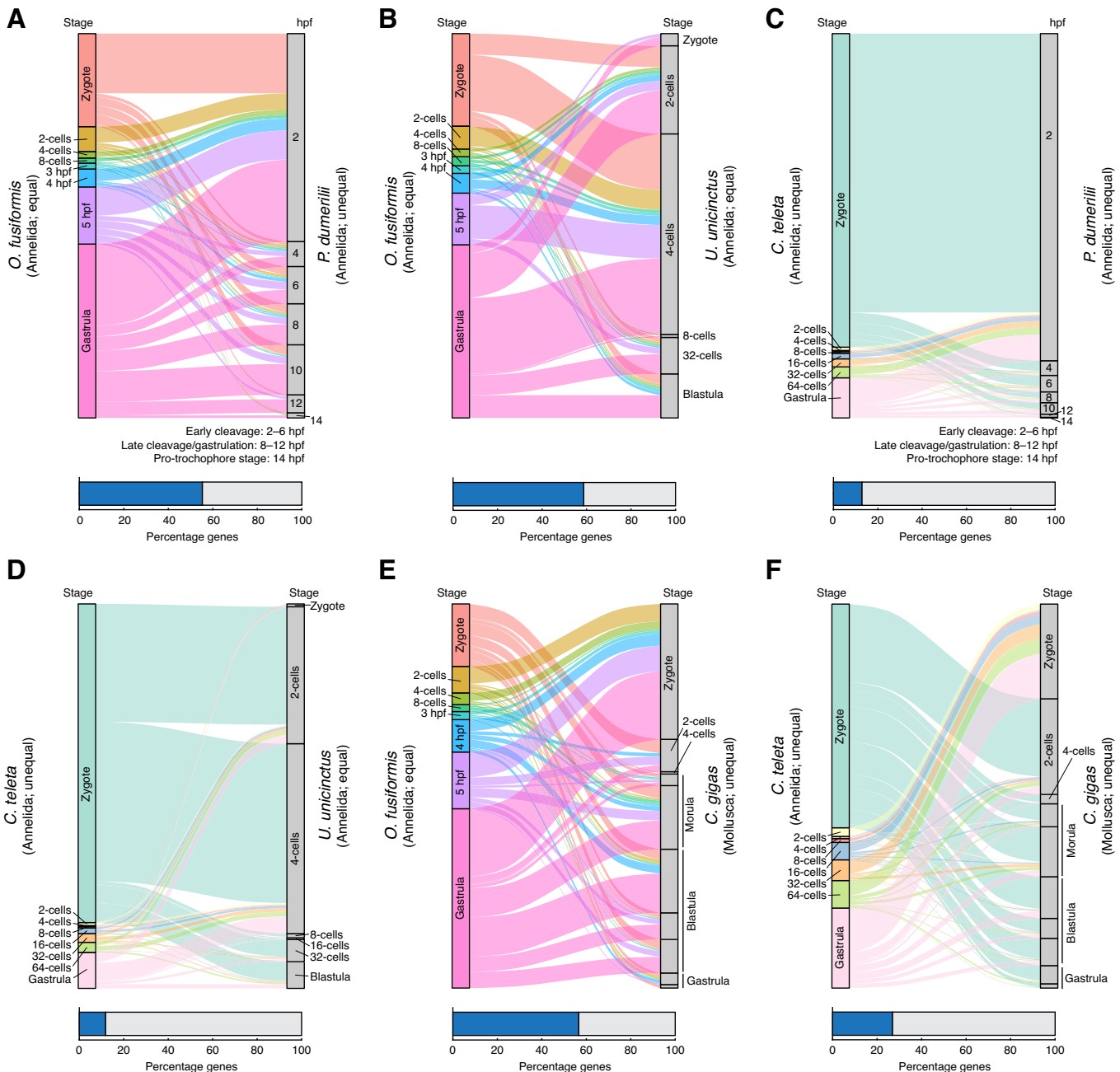

Early cleavage: 2–6 hpf
Late cleavage/gastrulation: 8–12 hpf
Pro-trochophore stage: 14 hpf

Early cleavage: 2–6 hpf
Late cleavage/gastrulation: 8–12 hpf
Pro-trochophore stage: 14 hpf

■ Percentage of genes expressed in the blastula and gastrula of *O. fusiformis* or *C. teleta* pre-displaced to earlier developmental stages in the respective comparison

**Figure EV4. Heterochronic shifts in gene expression between selected molluscan and annelid species.**

(A–F) Alluvial plots depicting the comparative deployment of one-to-one orthologs exhibiting shifts in temporal activation during spiral cleavage between *O. fusiformis* and *P. dumerilii* (A), *O. fusiformis* and *U. unicinctus* (B), *C. teleta* and *P. dumerilii* (C), *C. teleta* and *U. unicinctus* (D), *O. fusiformis* and *C. gigas* (E), and *C. teleta* and *C. gigas* (F). Generally, more genes shift from late cleavage stages in *O. fusiformis* to early stages in other species than when *C. teleta* is included in equivalent comparisons.

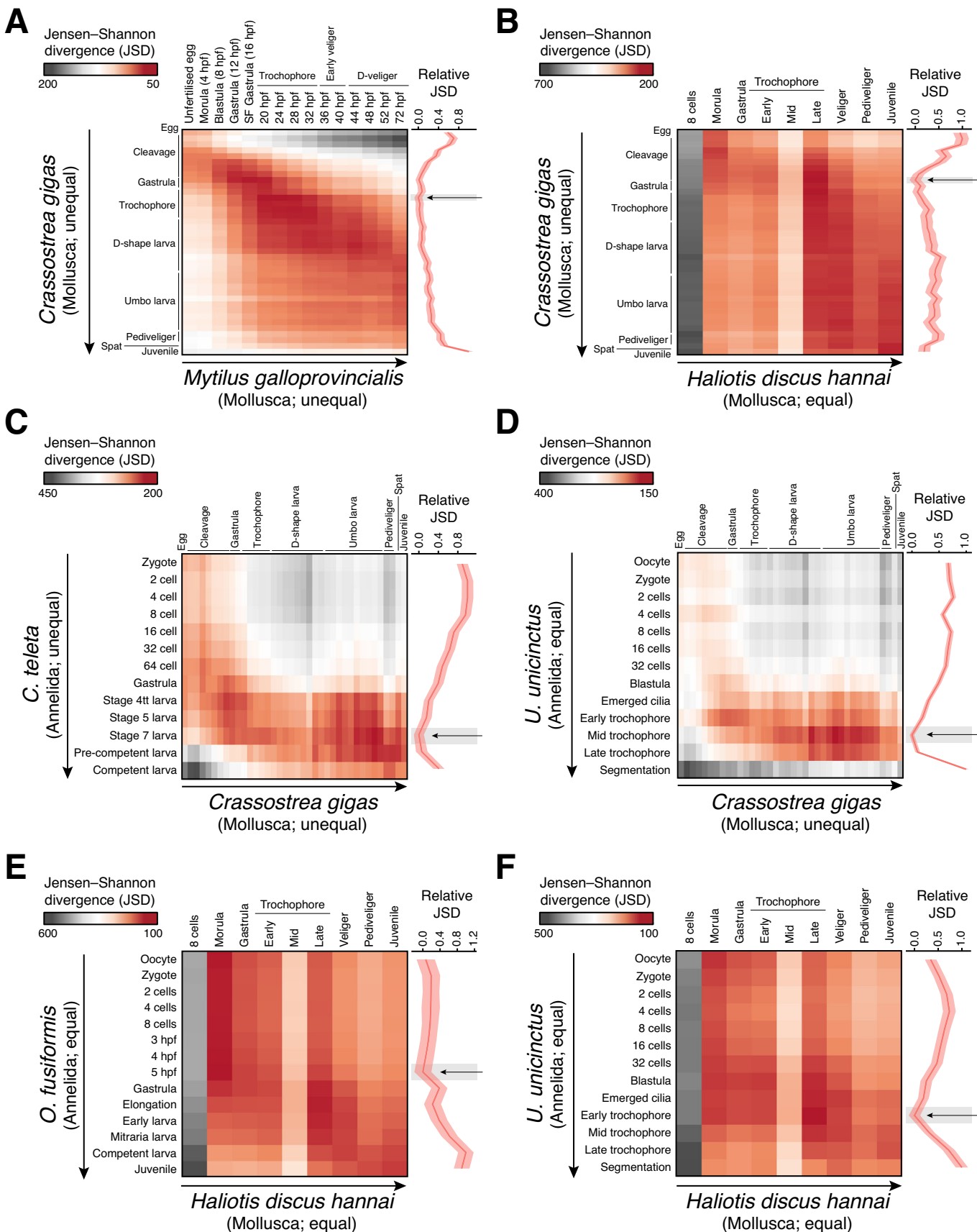

**Figure EV5.  Transcriptional similarity between Molluscan and Annelid development.**

(A–F) Jensen-Shannon transcriptomic divergence between all possible inter-species pairwise comparisons during the entire life cycle, from oocyte or cleavage to juvenile or competent larva, between molluscan and annelid species with a high-resolution time course. In intra-phylum comparisons, the stages of maximal similarity are at or around gastrulation. In contrast, in inter-phylum comparisons, the larval stages are more transcriptionally similar (except for the *O. fusiformis* versus *Haliotis discus hannai* comparisons, which might be due to the poor quality of the molluscan dataset).

