## [Peer Review File · EMBO Reports]

Cell fate specification modes shape transcriptome evolution in the highly conserved spiral cleavage

Yan Liang, Jingcheng Wei, Yue Kang, Allan Carrillo-Baltodano, and Jose Martin-Duran

Corresponding authors: Jose Martin-Duran (chema.martin@qmul.ac.uk)

Review Timeline:

Submission Date:	6th Jan 25
Editorial Decision:	11th Mar 25
Revision Received:	5th Jul 25
Editorial Decision:	23rd Jul 25
Revision Received:	30th Jul 25
Editorial Decision:	4th Aug 25
Revision Received:	5th Aug 25
Accepted:	12th Aug 25

Editor: Yehu Moran

Transaction Report:

Dear Dr. Martin-Duran

Thank you for the submission of your manuscript to EMBO reports. We have now received two sets of reviews for your paper that I include below.

As you will see, the referees acknowledge that the findings are potentially interesting. However, they do make significant comments that warrant your attention and should be address adequately.

I would thus like to invite you to revise your manuscript with the understanding that the referee concerns must be fully addressed and their suggestions taken on board. Please address all referee concerns in a complete point-by-point response. Acceptance of the manuscript will depend on a positive outcome of a second round of review. It is EMBO Reports policy to allow a single round of major revision only and acceptance or rejection of the manuscript will therefore depend on the completeness of your responses included in the next, final version of the manuscript.

The reason it took so long to get back to you with a decision letter is that we also invited a third referee that accepted the invitation but then did not return their review despite multiple reminders. Finally, this afternoon, we received via email the following message from them. Please try to accommodate this into your revision (I do understand it is not as helpful and structured as the two other reports):

"I think its a fine paper study..the questions are interesting and the analyses are well done. I think it massively over-states the conclusions...you cant make broad statements about how all annelids are, or how all spiralian are from a RNA-seq time series of two species. A two-taxon statement is flimsy in my opinion. I would ask them to dial it way back."

We realize that it is difficult to revise to a specific deadline. In the interest of protecting the conceptual advance provided by the work, we recommend a revision within 3 months (11th Jun 2025). Please discuss the revision progress ahead of this time with the editor if you require more time to complete the revisions.

- 1) A data availability section providing access to data deposited in public databases is missing. If you have not deposited any data, please add a sentence to the data availability section that explains that.
- 2) Your manuscript contains statistics and error bars based on $n=2$. Please use scatter blots in these cases. No statistics should be calculated if $n=2$.

5) a complete author checklist, which you can download from our author guidelines <<https://www.embopress.org/page/journal/14693178/authorguide>>. Please insert information in the checklist that is also reflected in the manuscript. The completed author checklist will also be part of the RPF.

6) Please note that all corresponding authors are required to supply an ORCID ID for their name upon submission of a revised manuscript (<<https://orcid.org/>>). Please find instructions on how to link your ORCID ID to your account in our manuscript tracking system in our Author guidelines <<https://www.embopress.org/page/journal/14693178/authorguide#authorshipguidelines>>

12) All Materials and Methods need to be described in the main text using our 'Structured Methods' format, which is required for all research articles. According to this format, the Methods section includes a Reagents and Tools Table (listing key reagents, experimental models, software and relevant equipment and including their sources and relevant identifiers) followed by a Methods and Protocols section describing the methods using a step-by-step protocol format. The aim is to facilitate adoption of

the methodologies across labs. More information on how to adhere to this format as well as a downloadable template (.docx) for the Reagents and Tools Table can be found in our author guidelines:
<https://www.embopress.org/page/journal/14693178/authorguide#structuredmethods>.

An example of a Method paper with Structured Methods can be found here: <https://www.embopress.org/doi/full/10.1038/s44320-024-00037-6#sec-4>

I look forward to seeing a revised form of your manuscript when it is ready.

Yours sincerely,

Yehu Moran
Editor
EMBO Reports

Referee #1:

In their paper, Liang et al. compare transcriptome dynamics during early developmental stages of two polychaetes, the homoquadrant, conditional developer *Owenia* and the heteroquadrant, autonomous developer *Capitella*. They show that the transcriptomes of the two species differ significantly at early stages but then converge towards gastrula before diverging again. Therefore, the authors argue that gastrula represents the annelid phylotypic stage. In my view, the most important finding of the paper is that there is a heterochronic shift in the expression of the orthologous genes to earlier developmental stages in the autonomously developing *Capitella* in comparison to *Owenia*. The paper is interesting and well-written, however, I have several problems with it, both conceptual and technical, which, in my opinion, need to be clarified before it can be published.

Major comments

1. The authors suggest that there is a gap between the disappearance of the maternal transcripts and the onset of zygotic expression. This seems extremely unusual to me. Normally, there is an overlap between maternal and zygotic transcripts, which does not seem to be the case in both studied polychaetes. Moreover, the authors seem to think that maternal mRNAs are not polyadenylated. They most certainly are, but in the maternal mRNA removal phase, one of the triggers for the degradation of the maternal transcripts is their de-adenylation. Since the authors used polyA-enrichment in their RNA-Seq library preparation, there is a distinct possibility that they lost a lot of maternal transcripts, which were being de-adenylated at the time when the authors see a dip in the read counts, and thus lost the overlap between maternal and zygotic transcription. In general, in my view, a more targeted analysis of maternal transcript dynamics (with the zygotic transcription blocked by actinomycin D or similar) vs. zygotic transcript dynamics (analysed by labelling and sequencing only nascent mRNA) is required to draw the conclusions the author want to make.

2. The authors use one annelid species with conditional development and one with autonomous development, but the manuscript reads as if their findings were generalizable to all annelids with the respective development types. Ideally, to make their conclusions more robust, the authors have to sequence several more annelids and molluscs with homo- and heteroquadrant development. This suggestion is, obviously, out of scope for the current paper, and I am, of course, not sufficiently familiar with the literature, but I would recommend that the authors include every annelid and mollusc model, for which RNA-Seq data are available at least for some of the stages. What about *Platynereis*, for example? The only non-annelid comparison was made against the oyster, and only on whole transcriptomes. In my opinion, it would be extremely useful to repeat the comparison also using orthologous genes, as it has been done for the *Owenia/Capitella* pair.

3. I was not at all surprised to see that, in the *Owenia*/oyster comparison, larval stages were the most similar at the level of transcriptomes (although I would increase the number of mollusc species if data are available). In contrast, the higher similarity of the gastrula transcriptomes in *Owenia* and *Capitella* than of their larval transcriptomes was very surprising. Will this hold if additional species were added into the comparison (again, *Platynereis*, but also other polychaetes)? How do you explain this rather unexpected result?

4. I think that the way the authors phrase one of their main messages is somewhat misleading. For example, already in the abstract they write "Together, our data reveal hidden developmental plasticity in the genetics underpinning spiral cleavage, indicating an evolutionary decoupling of morphological and transcriptomic conservation during early animal embryogenesis." I do not think that it is possible to conclude that the transcriptomic changes Liang et al. find are the "underpinnings of the spiral cleavage". In my opinion, they showed that spiral cleavage does not care at all about the transcriptome dynamics but rather proceeds under control of the yet unknown regulators, which are first deposited maternally and then, most likely, expressed non-differentially in the autonomous and conditional modes of development. In contrast, the authors showed, that the mode of development clearly affects the transcriptomes, which is expected, but still interesting. It is important that the authors do not try to "sell" their results as the genetic underpinnings of the spiral cleavage at any point of the manuscript.

Minor comments

Lines 51-52: Misleading sentence. Larval skeleton is not a unique trait of sea urchins. Brittle stars also have larval skeleton developing from the ingressed skeletogenic mesenchyme cells. The only difference is that skeletogenic mesenchyme does not form from micromeres.

Lines 85-86: In the text above, you mention only two modes (homoquadrant cleavage and induction vs. heteroquadrant cleavage and determination. But then you are just using plural. Are there other modes?

Lines 109-110: If RNA-Seq was performed at every cell division in *Owenia* and *Capitella*, please use cell numbers in both cases rather than hpf in *Owenia* and cell numbers in *Capitella*. Otherwise it is very confusing for a non-specialist like myself. If making stage labeling fully comparable is not possible because this reflects an unfortunate difference in material sampling strategy, this needs to be explicitly described in the main text as a limitation of your comparison. In that case, you should also provide an estimated cell number in the *Owenia* embryo at each particular hpf.

Also, in Fig. 6, you are using transcriptomes from later developmental stages, which is great, but I do not find the description of when and how RNA from these stages was sampled or where these data came from.

Lines 383-385: You show that in *Capitella*, RNA of these TFs appears earlier but broadly. What could be the function at this stage? Is there any reason to have these mRNAs there?

Referee #2:

The article is focused on the early embryogenesis of two annelid species, *Owenia fusiformis* and *Capitella teleta*, specifically on their spiral cleavage, which the authors claim to be a highly conserved developmental program. By means of high-resolution transcriptomic analyses, they show that their initial cleavage pattern is conserved, but their transcriptional dynamics differ significantly, influenced by the timing and mechanisms of embryonic organizer specification. They identify a "mid-developmental transition" during gastrulation where gene expression profiles and molecular patterning converge, before the species-specific divergence in larval forms.

The study suggests that, while the spiral cleavage program is highly conserved, it allows flexibility in transcriptional programs, which supports evolutionary diversity in developmental mechanisms. It provides valuable insights into how conserved embryonic processes adapt and evolve across species, and will be very interesting for the EvoDevo community.

In my opinion, this study would benefit from epigenomic experiments, which could complement the transcriptomic results and help infer relationships between transcription factors (and even reconstruct and compare GRNs). However, I understand that sample preparation in these species is not trivial, and the amount of available material may also be limiting. Overall, the analyses and experiments are well designed, the manuscript is clear and well written, and the conclusions are strongly supported by the data. I really enjoyed reading it. For these reasons, I would like to congratulate the authors on their excellent work and endorse the article for publication in EMBO Reports.

Response to Reviewers

We want to thank the two referees for their positive and constructive comments, which have significantly enhanced our work. We are pleased to provide a revised manuscript that addresses the comments raised. The main changes to our manuscript are summarised as follows:

Following the referee's #1 suggestion, we have expanded our investigations on the timings and dynamics of the maternal-to-zygotic transitions in *Owenia fusiformis* and *Capitella teleta*. We now include two additional validations: **(i)** a time-course series of immunolabelling against RNA polymerase II, and **(ii)** actinomycin D treatments in both species. These experiments demonstrate the nuclearisation of RNA pol II at the 4-cell stage and reveal a critical window of zygotic transcription between the 8- and 16-cell stages in both species. Together, they support our original conclusions based on RNA-seq data and provide a more comprehensive view of the zygotic genome activation in these annelids. In addition, we have included RNA-seq time-course data for five new species (the annelids *Platynereis dumerilii*, *Streblospio benedictii* and *Urechis unicinctus*, and the molluscs *Mytilus galloprovincialis*, and *Halyotis discus hannai*) in our genome-wide transcriptomic comparisons. These now strengthen our previous observations of the disparate transcriptomic dynamics during spiral cleavage and the inter- and intra-phylum similarities of the gastrula and larva stages in Spiralia, respectively. Finally, we have adjusted the tone of some sections of our manuscript and included all other minor text and figure suggestions.

Below, we provide a detailed point-by-point response to each referee's concerns. We hope we have addressed them satisfactorily and will be happy to address any additional comments or suggestions.

Referee #1:

In their paper, Liang et al. compare transcriptome dynamics during early developmental stages of two polychaetes, the homoquadrant, conditional developer Owenia and the heteroquadrant, autonomous developer Capitella. They show that the transcriptomes of the two species differ significantly at early stages but then converge towards gastrula before diverging again. Therefore, the authors argue that gastrula represents the annelid phylotypic stage. In my view, the most important finding of the paper is that there is a heterochronic shift in the expression of the orthologous genes to earlier developmental stages in the autonomously developing Capitella in comparison to Owenia. The paper is interesting and well-written, however, I have several problems with it, both conceptual and technical, which, in my opinion, need to be clarified before it can be published.

RESPONSE: Many thanks for the positive appraisal.

Major comments

1. *The authors suggest that there is a gap between the disappearance of the maternal transcripts and the onset of zygotic expression. This seems extremely unusual to me. Normally, there is an overlap between maternal and zygotic transcripts, which does not seem to be the case in both studied polychaetes. Moreover, the authors seem to think that maternal mRNAs are not polyadenylated. They most certainly are, but in the maternal mRNA removal phase, one of the triggers for the degradation of the maternal transcripts is their deadenylation. Since the authors used polyA-enrichment in their RNA-Seq library preparation, there is a distinct possibility that they lost a lot of maternal transcripts, which were being deadenylated at the time when the authors see a dip in the read counts, and thus lost the overlap between maternal and zygotic transcription. In general, in my view, a more targeted analysis of maternal transcript dynamics (with the zygotic transcription blocked by*

actinomycin D or similar) vs. zygotic transcript dynamics (analysed by labelling and sequencing only nascent mRNA) is required to draw the conclusions the author want to make.

RESPONSE: Following the referee's suggestion, we have defined more robustly when zygotic transcription starts and is essential for normal embryogenesis. Unfortunately, repeated efforts to implement EU-labelling to detect nascent mRNAs in these embryos at early stages were unsuccessful. As an alternative, we performed an analysis of RNA pol II nuclearisation from the 2-cells to the 32-cells stages in both species, demonstrating that RNA pol II becomes nuclear at the 4-cells stage in the two annelids, which is consistent with the increase in relative transcriptional levels for early zygotic genes as observed in Figure 4B, C. Notably, the nuclearisation at the 4-cell stage is homogenous in *O. fusiformis* (equal/homoquadrant) but asymmetric in *C. teleta* (unequal/heteroquadrant), with the C and D blastomeres showing more intense signal than A and B. To complement these findings, we performed a time-course analysis of transcriptional activation during early cleavage. In both annelids, we treated embryos with actinomycin D from the 2-cell stage to the 4-, 8-, 16- and 32-cell stages (in both species) and 5 and 6 hpf (in *O. fusiformis*). In the two species, blocking zygotic transcription during the 4- and 8-cell stages does not cause major developmental defects, while inhibiting transcription until and beyond the 16-cell stage compromises normal embryogenesis. This is consistent with our initial observation of increased differentially expressed genes at the 16-cell stage in *O. fusiformis* and *C. teleta*. Interestingly, and in support of our original data, the inhibition of zygotic transcription from 3 to 5 hpf affects embryogenesis but does not fully inhibit cell differentiation (e.g., ciliated cells are present) in *O. fusiformis*. Only inhibiting transcription until and beyond 5 hpf completely altered development, which is consistent with the large increase in differential gene expression associated with the specification of the embryonic organiser at 5 hpf in this species. Together, these new data qualify our previous model, indicating that the zygotic genome activation and maternal-to-zygotic transition are probably a longer process that starts around the 4-cell stage in the two annelids and culminates around the 16-cell stage, when zygotic transcription is already essential for normal development. This also supports our model that maternal transcripts decay significantly around the 16-cell stage (Figure 4B, C), as they cannot "rescue" the lack of zygotic transcription after actinomycin D treatment. In the future, optimisation of labelling and sequencing of nascent RNAs and RNA-seq ribodepletion (to avoid de-adenylation of maternal transcripts) will provide a more detailed perspective on the exact gene-specific dynamics of maternal and zygotic transcripts during the maternal-to-zygotic transition. However, including these datasets is beyond this manuscript's goal and the revision's time frame.

We have now included all these new datasets in two new figures (Figure 3A, B and Figure EV1) and a new paragraph in the Results section (lines 177–200). We have amended the discussion and final figure, accordingly, explicitly indicate that maternal and zygotic transcripts likely overlap during the 4- to 16-/32-cell stage, and highlighted the technical limitations of using poly-A selection to fully characterise maternal transcripts that are probably de-adenylated before degradation (lines 507–522).

2. The authors use one annelid species with conditional development and one with autonomous development, but the manuscript reads as if their findings were generalizable to all annelids with the respective development types. Ideally, to make their conclusions more robust, the authors have to sequence several more annelids and molluscs with homo- and heteroquadrant development. This suggestion is, obviously, out of scope for the current paper, and I am, of course, not sufficiently familiar with the literature, but I would recommend that the authors include every annelid and mollusc model, for which RNA-Seq data are available at least for some of the stages. What about Platynereis, for example? The only non-annelid

comparison was made against the oyster, and only on whole transcriptomes. In my opinion, it would be extremely useful to repeat the comparison also using orthologous genes, as it has been done for the *Owenia/Capitella* pair.

3. I was not at all surprised to see that, in the *Owenia*/oyster comparison, larval stages were the most similar at the level of transcriptomes (although I would increase the number of mollusc species if data are available). In contrast, the higher similarity of the gastrula transcriptomes in *Owenia* and *Capitella* than of their larval transcriptomes was very surprising. Will this hold if additional species were added into the comparison (again, *Platynereis*, but also other polychaetes)? How do you explain this rather unexpected result?

RESPONSE (to points 2 and 3): We have now expanded our transcriptomic comparisons to five other spiralian species for which transcriptomic data for several developmental time points are available: the annelids *Platynereis dumerilii* (as the referee suggested), *Urechis unicinctus* and *Streblospio benedicti* (including both lecithotrophic and planktotrophic larval types), and the molluscs *Mytilus galloprovincialis* and *Haliotis discus hannai*. These new species cover a diversity of developmental strategies (*U. unicinctus* and *H. discus hannai* are equal/conditional developers, and *P. dumerilii*, *S. benedicti* and *M. galloprovincialis* are unequal/autonomous developers) and phylogenetic groups (e.g., *U. unicinctus* belongs to the sister annelid group to *C. teleta*; *M. galloprovincialis* and *C. gigas* are both bivalves, while *H. discus hannai* is a gastropod). Although, as expected, these datasets exhibit technical (e.g., sequencing mode and depth, number of replicates) and sampling differences (e.g., some only include an early and a late cleavage stage) that affect the comparisons, the overall analyses support our original conclusions. In particular, adding the three new annelids confirms the distinct transcriptomic dynamics during spiral cleavage and the convergence at the gastrula stage in Annelida, in line with the referee's point #4. The analysis of orthologous genes (referee's point 2) undergoing heterochronic shifts between these species also demonstrate that more genes tend to shift from late to early stages between *O. fusiformis* and other spiralian than between *C. teleta* and other species (in this analysis, *S. benedicti* and *H. discus hannai* datasets were excluded because they only include one cleavage stage [8- or 16-cells], and *M. galloprovincialis* was excluded because it is redundant with *C. gigas* [both are closely related bivalves]). Finally, we compared the entire time courses of development (including post-gastrula stages) between annelids and molluscs (excluding *P. dumerilii* and *S. benedictii*, as they do not include late larval stages) to strengthen our initial observations that the highest transcriptomic similarity within phylum occurs early in the life cycle, around the gastrula stage. At the same time, the larvae are the most transcriptionally similar stage between the two phyla. The comparison between *O. fusiformis* and *H. discus hannai* is the only one diverging from this observation, but it might be due to technical issues with the *H. discus hannai* (note the oddly behaving 8-cell and mid trochophore samples in Supplementary Figure 18E, F).

We report these new analyses in four new supplementary figures (Figure EV2, Figure EV4, Figure EV5 and Appendix Figure S13) and at several points in the text (lines 252–254, 368–370, 384–387 and 453–458).

4. I think that the way the authors phrase one of their main messages is somewhat misleading. For example, already in the abstract they write "Together, our data reveal hidden developmental plasticity in the genetics underpinning spiral cleavage, indicating an evolutionary decoupling of morphological and transcriptomic conservation during early animal embryogenesis." I do not think that it is possible to conclude that the transcriptomic changes Liang et al. find are the "underpinnings of the spiral cleavage". In my opinion, they showed that spiral cleavage does not care at all about the transcriptome dynamics but rather proceeds under control of the yet unknown regulators, which are first deposited maternally

and then, most likely, expressed non-differentially in the autonomous and conditional modes of development. In contrast, the authors showed, that the mode of development clearly affects the transcriptomes, which is expected, but still interesting. It is important that the authors do not try to "sell" their results as the genetic underpinnings of the spiral cleavage at any point of the manuscript.

RESPONSE: Following the referee's suggestion, we have amended the last sentence of the abstract (lines 32–34: "Together, our data reveal hidden transcriptomic plasticity during spiral cleavage, indicating an evolutionary decoupling of morphological and transcriptomic conservation during early embryogenesis."). We have also adjusted other parts of the text to avoid overstating the functional implications of our data in explaining the genetic control of spiral cleavage (e.g., lines 109–112, lines 578–583).

Minor comments

Lines 51-52: Misleading sentence. Larval skeleton is not a unique trait of sea urchins. Brittle stars also have larval skeleton developing from the ingressed skeletogenic mesenchyme cells. The only difference is that skeletogenic mesenchyme does not form from micromeres.

RESPONSE: We have amended the sentence as follows: "For example, although sea stars and sea urchins share holoblastic radial cleavage, only the latter form their larval skeleton from a set of vegetal micromeres, a unique trait of this echinoderm lineage (Emura & Yajima, 2022)." (lines 52–55).

Lines 85-86: In the text above, you mention only two modes (homoquadrant cleavage and induction vs. heteroquadrant cleavage and determination. But then you are just using plural. Are there other modes?

RESPONSE: We have clarified these sentences and refer only to inductive (conditional) vs determinative (autonomous) modes (line 83).

Lines 109-110: If RNA-Seq was performed at every cell division in Owenia and Capitella, please use cell numbers in both cases rather than hpf in Owenia and cell numbers in Capitella. Otherwise it is very confusing for a non-specialist like myself. If making stage labeling fully comparable is not possible because this reflects an unfortunate difference in material sampling strategy, this needs to be explicitly described in the main text as a limitation of your comparison. In that case, you should also provide an estimated cell number in the Owenia embryo at each particular hpf.

RESPONSE: The embryos of *O. fusiformis* are small, and counting the number of cells under a stereomicroscope after the 8-cell stage is challenging. However, a previous study demonstrated that 3, 4, and 5 hours post-fertilisation time points correspond to the 16-, 32- and 64-cell stages, with only minimal asynchronies between embryos (Carrillo-Baltodano et al. 2021, *EvoDevo*). This unavoidable sampling difference is unlikely to affect the comparison and is now clarified in the text (lines 119–122) and figure 1. However, we continued to use the 3, 4, and 5 hpf nomenclature throughout the manuscript as it better describes our dataset.

Also, in Fig. 6, you are using transcriptomes from later developmental stages, which is great, but I do not find the description of when and how RNA from these stages was sampled or where these data came from.

RESPONSE: We now clarify this point in the Materials and Methods section (lines 710–715) and refer to a new supplementary table (Table S47) collecting all the datasets, stages and corresponding accession numbers used in this study (including the new species used to address points 2 and 3 above).

Lines 383-385: You show that in Capitella, RNA of these TFs appears earlier but broadly. What could be the function at this stage? Is there any reason to have these mRNAs there?

RESPONSE: The broad expression of the assessed TFs was unexpected. One possibility is that a late-cleavage/pre-gastrulation event restricts their expression to the progenitors of the areas showing expression of these genes at the gastrula stage, as we know it occurs for some genes in *O. fusiformis* (e.g., *Nodal* is broadly expressed in the gastral plate in the blastula and then on a single asymmetric cell in the gastrula; Carrillo-Baltodano et al. 2025, BioRxiv). However, this is speculative and more detailed analyses are required to resolve the exact developmental roles of these TFs in *C. teleta* early embryogenesis. We now comment on this finding in lines 423–424 (“The early broad expression was unexpected and warrants further investigation, for example, through gene-specific knockouts (Neal et al, 2019).”).

Referee #2:

The article is focused on the early embryogenesis of two annelid species, Owenia fusiformis and Capitella teleta, specifically on their spiral cleavage, which the authors claim to be a highly conserved developmental program. By means of high-resolution transcriptomic analyses, they show that their initial cleavage pattern is conserved, but their transcriptional dynamics differ significantly, influenced by the timing and mechanisms of embryonic organizer specification. They identify a "mid-developmental transition" during gastrulation where gene expression profiles and molecular patterning converge, before the species-specific divergence in larval forms.

The study suggests that, while the spiral cleavage program is highly conserved, it allows flexibility in transcriptional programs, which supports evolutionary diversity in developmental mechanisms. It provides valuable insights into how conserved embryonic processes adapt and evolve across species, and will be very interesting for the EvoDevo community.

In my opinion, this study would benefit from epigenomic experiments, which could complement the transcriptomic results and help infer relationships between transcription factors (and even reconstruct and compare GRNs). However, I understand that sample preparation in these species is not trivial, and the amount of available material may also be limiting. Overall, the analyses and experiments are well designed, the manuscript is clear and well written, and the conclusions are strongly supported by the data. I really enjoyed reading it. For these reasons, I would like to congratulate the authors on their excellent work and endorse the article for publication in EMBO Reports.

RESPONSE: Many thanks for the positive endorsement of our study. As the referee highlights, epigenomic data would be an insightful but non-trivial addition. We have included a sentence indicating that, in the future, epigenomic datasets will be an important addition to complement the transcriptomic data of our study (lines 638–642: “In the future, exploiting low-input epigenomic profile approaches, such as ATAC-seq and CUT&Tag (Buenrostro et al, 2015; Kaya-Okur et al, 2019), will complement our transcriptomic resources and help infer the functional and regulatory dynamics underpinning the differences in gene expression between species with autonomous and conditional spiral cleavage.”).

Dear Dr. Martin-Duran

Thank you for the submission of your revised manuscript to our offices. We have now received the enclosed reports from the referees that were asked to assess it. EMBOR-2025-61121V2 still has somewhat minor suggestions that I would like you to incorporate before we can proceed with the official acceptance of your manuscript. I will review your edits myself, but may still seek some quick advice from Reviewer #1 as they are an expert in this field.

Another important point is addressing the comments made by editorial assistance team that look more into the technical aspects of your paper. I provide their comments below. This is extremely important for facilitating a quick process for your manuscript as we would not be able to formally accept it until these points are sorted out.

Please do not hesitate to contact me at my email y.moran@emboreports.org should you have any questions regarding the revision process.

I look forward to seeing a new revised version of your manuscript as soon as possible.

Best regards,
Yehu Moran
Academic Editor
EMBO Reports

Comments by Editorial Assistant:

Conflict of Interest statement: in but needs to be placed after Acknowledgments

AC/CRedit: needs to be removed from the manuscript text and be included only via the system.

FUNDING INFO: Not congruent between the system and the text; missing in the system: BBSRC grant BB/W019698/1. Please correct.

FIGURE CALLOUTS: callouts of Appendix tables (and datasets) need to be updated to Appendix Table S1, etc. and Dataset EV1, etc.

DATASET EV LEGENDS: an Excel file with Appendix tables has been uploaded as Dataset 1 - the small tables that can fit the Appendix file should be removed and provided in the Appendix File as Appendix Table S# and their legend should be provided above the table; all the other complex tables with lots of rows (and columns) should be provided individually as separate dataset files: Dataset EV1, etc. the legends of each dataset should be provided in the same Excel file as a separate sheet/tab - the nomenclature needs to be corrected in all places (callouts included).

APPENDIX FILE WITH Table of Contents: included, but "Data" from the title needs to be removed, we need just "Appendix"; the Table of Contents (ToC) needs to have each figure and its page in the file listed on the title page; as for the tables, they have been provided in an Excel file, but the ones that can fit the Appendix file should be provided there and each should be listed with its page number in the Table of Contents while each legend should be provided above its table; the correct nomenclature of the Appendix items is Appendix Figure S1, etc and Appendix Table S1, etc. - the nomenclature needs to be corrected in all places (callouts included)

SYNOPSIS IMAGE: missing, please provide

SYNOPSIS TEXT: missing, please provide

R&T TABLE: in the manuscript and uploaded separately; the one in the manuscript needs to be removed

SOURCE DATA (SD): The following SD have been uploaded to the system: Figure 1B and Figure 3A; according to the checklist these should be deposited online, please check: 1CEF, 3DEF, 6ABCDEFGHIJKLMN; the SD in our system need to be provided as one zip folder per figure.

Additional notes:

- Materials and Methods should be Methods

Figure Legends - Comments

- Please note that the legend for figure 1, EV1 is not provided in the sequential manner. This needs to be rectified.

- Please indicate the statistical test used for data analysis in the legends of figures 3C, D; 4D

- Please note that the asterisk is not defined in the legend of figure 3E, F; 6A-G. This needs to be rectified.

Referee #1:

The revised version of the manuscript of Liang et al. is a significant improvement over the first version. I am happy to

recommend publication after minor textual changes, and I do not see the need for an additional revision round.

The only more or less serious change I would suggest is about the EV4 and EV5. While the results of the new actinomycin experiment are presented in the main text figure and well-described, the analysis of the author's exciting new interspecies comparisons is more or less missing. There is a single sentence in the lines 368-370 and another sentence in the lines 453-455. I would like to ask the authors to elaborate on this rather than simply state that other comparisons are in line with the *Owenia/Capitella* or *Owenia/oyster* comparisons and really explain what they see. If space is a problem, the parts on GO-term analysis and the phylostratigraphy can be shortened.

Minor comments:

1. Line 606: The authors write "Our data highlights the gastrula as a critical developmental time point in spiral cleavage". This must be a mistake. Cleavage ends when the cell cycle switches for alternating between the S- and the M-phase to incorporating G-phases. In all cases known to me, gastrulation takes place long after the end of cleavage. Please rephrase to something like "gastrula is a critical developmental time point in the spiralian embryos we studied".

2. A related point: I would recommend removing the slightly misleading explanation that mid-blastula transition is the desynchronization of the cell cycles (lines 496-497). Certainly, desynchronization of the cell cycles is one of the most noticeable manifestations of the MBT, but this desynchronization happens due to the zygotic genome activation and incorporation of the G1 and G2 phases into the cell cycle, which are the defining features of the MBT. In many animals, maternal-to-zygotic transition happens earlier than the blastula stage, hence, the term MBT fell out of use. The authors do not need the term MBT to convey their message. They use it only twice (in lines 497, and 518), and I would advise to stick to more rigorous terms "zygotic genome activation" or "maternal-to-zygotic transition", and "desynchronization of cell division".

3. On Fig. EV5C, *C. teleta* is labelled as Mollusca. Please correct.

Thanks for an interesting paper.

Referee #2:

I did not have any big concern with the previous version, and neither do I with this one, which indeed has been improved. I think the manuscript deserves publication in EMBO Reports.

Response to Reviewers

We want to thank the two referees for their positive and constructive comments. Below, we provide a detailed point-by-point response. We hope we have addressed them satisfactorily and will be happy to address any additional comments or suggestions you may have.

Referee #1:

The revised version of the manuscript of Liang et al. is a significant improvement over the first version. I am happy to recommend publication after minor textual changes, and I do not see the need for an additional revision round.

The only more or less serious change I would suggest is about the EV4 and EV5. While the results of the new actinomycin experiment are presented in the main text figure and well-described, the analysis of the author's exciting new interspecies comparisons is more or less missing. There is a single sentence in the lines 368-370 and another sentence in the lines 453-455. I would like to ask the authors to elaborate on this rather than simply state that other comparisons are in line with the Owenia/Capitella or Owenia/oyster comparisons and really explain what they see. If space is a problem, the parts on GO-term analysis and the phylostratigraphy can be shortened.

RESPONSE: Following the referee's suggestion, we have now expanded these parts of the manuscript. We have written a new paragraph on Figure EV4 (heterochronic shifts between annelids and molluscs; lines 390–414) and expanded our description of Figure EV5 (genome-wide transcriptomic comparisons) into another paragraph (lines 478–489). We hope this new text provides a more elaborate description of the interspecies, interphyla comparisons.

Minor comments:

1. Line 606: The authors write "Our data highlights the gastrula as a critical developmental time point in spiral cleavage". This must be a mistake. Cleavage ends when the cell cycle switches for alternating between the S- and the M-phase to incorporating G-phases. In all cases known to me, gastrulation takes place long after the end of cleavage. Please rephrase to something like "gastrula is a critical developmental time point in the spiralian embryos we studied".

RESPONSE: Modified as suggested (line 606).

2. A related point: I would recommend removing the slightly misleading explanation that mid-blastula transition is the desynchronization of the cell cycles (lines 496-497). Certainly, desynchronization of the cell cycles is one of the most noticeable manifestations of the MBT, but this desynchronization happens due to the zygotic genome activation and incorporation of the G1 and G2 phases into the cell cycle, which are the defining features of the MBT. In many animals, maternal-to-zygotic transition happens earlier than the blastula stage, hence, the term MBT fell out of use. The authors do not need the term MBT to convey their message. They use it only twice (in lines 497, and 518), and I would advise to stick to more rigorous terms "zygotic genome activation" or "maternal-to-zygotic transition", and "desynchronization of cell division".

RESPONSE: We have removed the term “mid-blastula transition” in line 497 and replaced it with “desynchronisation of cell divisions” in line 518.

3. On Fig. EV5C, C. teleta is labelled as Mollusca. Please correct.

RESPONSE: Corrected

Thanks for an interesting paper.

RESPONSE: Many thanks for your time and constructive criticism.

Referee #2:

I did not have any big concern with the previous version, and neither do I with this one, which indeed has been improved. I think the manuscript deserves publication in EMBO Reports.

RESPONSE: Many thanks.

Manuscript number: EMBOR-2025-61121V3

Title: Cell fate specification modes shape transcriptome evolution in the highly conserved spiral cleavage

Author(s): Yan Liang, Jingcheng Wei, Yue Kang, Allan Carrillo-Baltodano, and Jose Martin-Duran

Dear Dr. Martin-Duran,

Thank you for your patience while we have reviewed your revised manuscript. I am writing with an 'accept in principle' decision, which means that I will be happy to accept your manuscript for publication once a last minor issue/correction has been addressed, as follows: this might seem very small, maybe even a petty request, but we do try to keep a certain style in our journal as we believe it is important in order to provide our readership with added value when they access our website: our synopsis image for each article is intended to provide the readers with information, and not just be a nice picture. Please take a look on our website for examples and resubmit a graphic that is more informative for your paper as a synopsis image. If you have some doubts or questions what can be a suitable image please do not hesitate to consult with me at y.moran@emboreports.org

Once you have made this (very) minor revision, please use the following link to submit your corrected manuscript:

Link Not Available

When all remaining corrections have been attended to, you will then receive an official decision letter from the journal accepting your manuscript for publication in the next available issue of EMBO Reports. This letter will also include details of the further steps you need to take for the prompt inclusion of your manuscript in our next available issue.

Thank you for your contribution to EMBO Reports.

Yours sincerely,

Yehu Moran
Academic Editor
EMBO Reports

The authors addressed the remaining editorial issues.

Dr. Jose Martin-Duran
Queen Mary University of London
Mile End Road
Fogg Building
London E1 4NS
United Kingdom

Dear Dr. Martin-Duran,

I am very pleased to accept your manuscript for publication in the next available issue of EMBO Reports. Thank you for your contribution to our journal.

Yours sincerely,

Yehu Moran
Academic Editor
EMBO Reports
